# Bridging Radiology and Pathology Foundation Models via Concept-Based Multimodal Co-Adaptation

**Yihang Chen**[1], **Yanyan Huang**[1], **Fuying Wang**[2], **Maximus Yeung**[1],
**Yuming Jiang**[3], **Shujun Wang**[4], and **Lequan Yu**[1]*
[1]University of Hong Kong      [2]Stanford University
[3]Wake Forest University School of Medicine      [4]The Hong Kong Polytechnic University
{yihangc, yanyanh}@connect.hku.hk
fuyingw@stanford.edu, yuming.jiang@wfusm.edu
shu-jun.wang@polyu.edu.hk, {mcfyeung, lqyu}@hku.hk

## Abstract

Pretrained medical foundation models (FMs) have shown strong generalization across diverse imaging tasks, such as disease classification in radiology and tumor grading in histopathology. While recent advances in parameter-efficient fine-tuning have enabled effective adaptation of FMs to downstream tasks, these approaches are typically designed for a single modality. In contrast, many clinical workflows rely on joint diagnosis from heterogeneous domains, such as radiology and pathology, where fully leveraging the representation capacity of multiple FMs remains an open challenge. To address this gap, we propose **C**oncept **T**uning and **F**using (**CTF**), a parameter-efficient framework that uses clinically grounded concepts as a shared semantic interface to enable cross-modal co-adaptation *before* fusion. By incorporating task-specific concepts that are relevant across modalities, CTF aligns radiology and pathology representations, thereby enhancing their complementarity and enabling interpretation. We further design a **G**lobal–**C**ontext–**S**hared **P**rompt (**GCSP**) mechanism, which employs a small set of learnable tokens to capture domain-specific priors, shared patient-level information, and cross-domain context. The resulting concept alignment scores from each modality are then fused to produce a final prediction. Extensive experiments demonstrate that CTF outperforms strong unimodal, latent-fusion, and adapter-based baselines (e.g., AUC 0.903 on TCGA-GBMLGG). Notably, CTF achieves these gains without finetuning the full FMs, requiring only 0.15% additional parameters, thus highlighting the effectiveness of concept-based multimodal co-adaptation. The code is available at `https://github.com/HKU-MedAI/CTF`.

## 1 Introduction

Foundation models (FMs) are increasingly demonstrating significant potential in transforming healthcare by enabling the joint analysis of medical images and associated textual information (Cui et al., 2023; Steyaert et al., 2023; Qian et al., 2021; Radford et al., 2021). In clinical practice, however, a patient's condition is often assessed through multiple diagnostic domains[1], such as radiology scans (e.g., CT, MRI) providing macroscopic structural information and pathology slides revealing microscopic cellular details (Tomaszewski & Gillies, 2021; Qi et al., 2024). Integrating information from these diverse sources is crucial for a holistic understanding of disease processes and accurate prediction of clinical outcomes like patient survival or tumor grade (Rahaman et al., 2025; Wang et al., 2024). Yet, many current vision-language FM applications in healthcare operate within siloed domains—one for radiology, another for pathology—each with its own "visual language"

---

*Corresponding author.
[1]We use the term "domain" to refer to pathology and radiology, and "modal" to refer to texts and images.

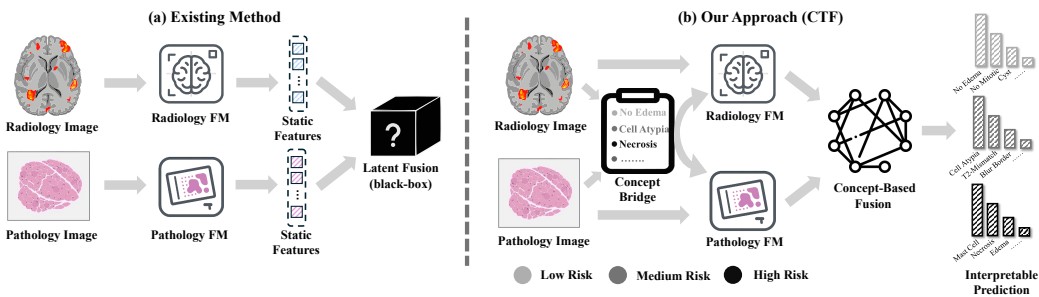

Figure 1: Conceptual comparison of multimodal fusion paradigms. (a) Conventional pipeline: radiology and pathology FMs are used as frozen feature extractors; fusion is performed on static latent features. (b) CTF performs cross-domain co-adaptation of risk-stratified concept semantics before fusion, enabling interpretable predictions.

(Lu et al., 2024; Zhang et al., 2023; Majzoub et al., 2025). The central challenge, therefore, is not merely to combine their outputs but to bridge these distinct expert models in a way that creates deep, synergistic understanding and maintains the clinical interpretability crucial for high-stakes medical decisions.

Existing attempts at multi-domain integration in medicine face practical and methodological limits. A common strategy uses pre-trained Vision Language Models (VLMs) as fixed feature extractors for each domain, followed by a simple fusion mechanism (e.g., concatenation) applied to these static representations (Zhang et al., 2024a; Xu & Chen, 2023). This approach limits the model's adaptability to the specific information of the downstream task and the interplay between modalities. Conversely, full fine-tuning of large VLMs is computationally expensive and often remains confined to the model's pretraining domain, weakening knowledge transfer across domains (Shi et al., 2024). Furthermore, both approaches often result in "black-box" systems where the reasoning behind predictions remains opaque, posing challenges for clinical trust and interpretability—a critical requirement in high-stakes medical decision-making (Amann et al., 2020; Rudin, 2019; Doshi-Velez & Kim, 2017). This conventional paradigm, which relies on extracting static features from siloed modalities before a simple, non-interpretable fusion step, is illustrated in Figure 1a.

To overcome these challenges, we argue that the key to unlocking synergy is to establish a shared, interpretable vocabulary that can bridge the semantic gap between FMs from different domains. Clinically-grounded concepts, such as "tumor necrosis" or "cellular atypia", provide this natural bridge. However, treating concepts as fixed definitions is brittle: the prognostic meaning of a concept in one domain often depends on context from the other. For example, "irregular tumor margins" in radiology is far more alarming when paired with histopathology evidence of "lymphovascular invasion." While recent works have begun aligning expert-derived concepts with images (Zhao et al., 2024; Nguyen et al., 2025; Zhu et al., 2025a), they do not dynamically modulate these concepts based on cross-domain information. Our core insight is that concepts should not be a static bottleneck, but a dynamic medium for co-adaptation, where the semantic representation of a concept in one modality is actively tuned by features from the other.

To this end, we introduce **Concept Tuning and Fusing (CTF)**, a novel framework that bridges radiology and pathology VLMs through medically-enriched concepts (Figure 1b). Instead of fusing static features, CTF forces each modality to "be aware" of the other during the feature extraction process. The core of our framework is the Global-Context-Shared Prompt (GCSP) strategy, a prefix tuning method that conditions the interpretation of concepts within one domain (e.g., radiology) on the visual features from the complementary domain (e.g., pathology). This cross-domain conditioning allows each VLM to produce richer, contextually-aware concept representations tailored to the patient case before they are fused for a final prediction. We detail three stages: (i) MI + diversity concept selection (Sec 3.1), (ii) GCSP-based cross-domain concept tuning (Sec 3.2), and (iii) concept-score fusion and prediction (Sec 3.3).

Our main contributions are:

- We propose **CTF**, a new framework that uses medically-relevant concepts as a dynamic and interpretable bridge to fuse distinct medical VLMs, moving beyond conventional "black-box" latent fusion.

- We introduce the **Global-Context-Shared Prompt (GCSP)** strategy, a novel cross-domain conditioning mechanism that uses efficient prompt tuning ($\approx$0.15% trainable parameters) to adapt the semantic meaning of concepts based on complementary domains.

- We conduct extensive experiments on four public and in-house datasets, demonstrating that CTF significantly outperforms state-of-the-art unimodal and multimodal methods in both survival analysis and cancer grading, achieving a C-index improvement of 3.5% and an AUC improvement of 2.9% over the strongest baselines, respectively.

## 2 RELATED WORK

**Multimodal Fusion for Clinical Prediction.** The integration of diverse data, especially macroscopic radiology and microscopic pathology, is critical in oncology (Zhao et al., 2022; Zhu et al., 2025b), with multimodal models consistently outperforming unimodal approaches (Benani et al., 2025). A dominant paradigm is latent-space fusion, where features are independently extracted and then combined using methods like co-attention transformer (Xu & Chen, 2023; Lu et al., 2019; Chen et al., 2023), or information-theoretic disentanglement to separate shared and specific information (Zhang et al., 2024a). While powerful, these methods primarily fuse latent representations that are already fixed, treating the feature extractors as black boxes and limiting the depth of synergy. Even when powerful pathology foundation models are used as the pathology encoder (Chen et al., 2024a; Ma et al., 2025; Xu et al., 2024), they are typically plugged in as static feature extractors within this latent-fusion pipeline. Recent robust MIL methods for WSI analysis, such as cDP-MIL and SIB-MIL (Chen et al., 2024b; 2025a), further highlight the importance of handling instance-level noise and heterogeneity in pathology representations. Our work proposes a shift: instead of refining the fusion of static features, we enable a dynamic dialogue between domains. Unlike latent fusion that combines already-fixed features, and fine-tuning that adapts backbones separately, CTF explicitly co-adapts concept semantics using cross-domain prompts *before* any fusion.

**Foundation Model Adaptation and Cross-Domain Guidance.** The advent of foundation models (FMs) offers new avenues for realizing this dialogue. While many works use FMs as off-the-shelf feature extractors, Parameter-Efficient Fine-Tuning (PEFT) (Hu et al., 2022; Li & Liang, 2021) methods have emerged to adapt them. Techniques include architectural adapters (Houlsby et al., 2019) and recent medical adapters (Lee et al., 2025) or prompt-based tuning (Lester et al., 2021), which have been used to adapt VLMs for specific domains or tasks (Zhou et al., 2022), often in a unimodal fashion (Peng et al., 2025). In computational pathology, knowledge-enhanced compression and prompt-like adaptation have been explored for few-shot WSI classification (Guo et al., 2025), and parameter-efficient tuning has also been leveraged for medical report generation in federated settings (Che et al., 2025). CTF innovates by using prompts not just for task adaptation, but as a cross-domain conditioning mechanism where one domain dynamically influences the semantic interpretation within another, creating an integrated, context-aware system *before* fusion.

**Concept-Based and Interpretable Multimodal Learning.** A major barrier to the clinical adoption of deep fusion models is their lack of interpretability (Amann et al., 2020). Concept Bottleneck Models (CBMs) (Koh et al., 2020; Ghorbani et al., 2019) address this by forcing predictions through a set of human-understandable concepts. While foundational, this rigid bottleneck can limit performance. The field is evolving towards more flexible, interpretable frameworks, such as using multimodal contrastive learning (Nauta et al., 2021) to find local, explainable correlations between imaging and text or building large-scale, concept-centric FMs like ConceptCLIP (Nie et al., 2025). CTF is inspired by this philosophy but makes a crucial contribution to the multimodal context. We treat concepts not as a static bottleneck, but as the very medium for the cross-domain guidance described above. Our key innovation, the GCSP strategy, allows each VLM to adjust its understanding of concepts like "tumor invasiveness" based on real-time information from the complementary domain. This unique synthesis provides the deep synergy of cross-domain guidance while leveraging the transparency of concept-based reasoning, distinguishing our work from both black-box fusion techniques and traditional CBMs.

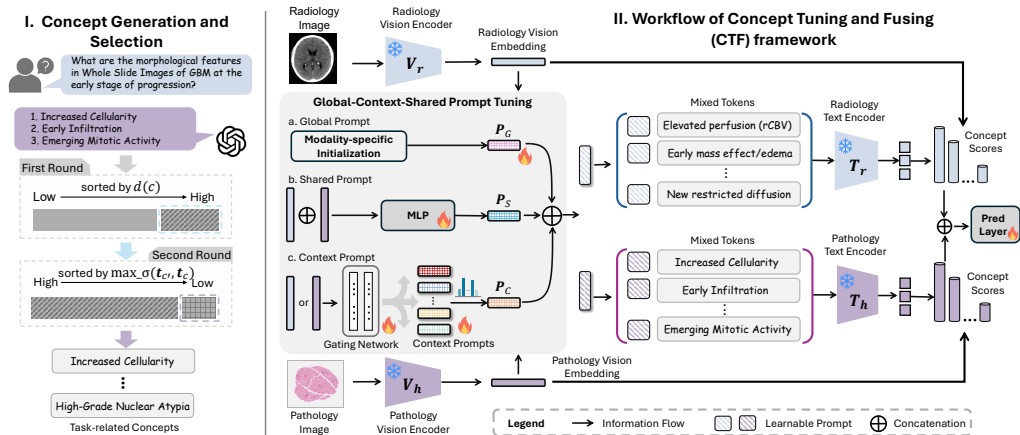

Figure 2: Overview of the CTF framework.(I) Concept Generation & Selection identifies relevant and diverse medical concepts. (II) Concept Tuning & Fusing freezes radiology/pathology vision encoders. GCSP prepends three prompts to each concept's text tokens—(a) global (task adaptation), (b) shared (MLP over $[\boldsymbol{f}_r, \boldsymbol{f}_h]$), and (c) context (gated by the complementary modality)—before the frozen text encoders, producing per-domain concept scores that are concatenated for prediction.

## 3 METHODOLOGY

Our Concept Tuning and Fusing (CTF) framework enables synergistic integration of radiology and pathology data by creating a shared semantic bridge built on medically relevant concepts. The framework, depicted in Figure 2, proceeds in three main stages. First, in **Prognostic Concept Selection (Sec 3.1)**, we generate a comprehensive pool of medical concepts and then use a principled optimization strategy to select a compact, diverse, and prognostically relevant subset for each domain. Second, in the core of our framework, **Cross-Domain Concept Co-Adaptation (Sec 3.2)**, we use our novel Global-Context-Shared Prompt (GCSP) mechanism to dynamically tune the textual embedding of each concept, making it aware of both the downstream task and the cross-domain context of the specific patient case. Finally, in **Fusion and Interpretable Prediction (Sec 3.3)**, these co-adapted concept scores are fused and fed to a prediction head, ensuring that the final output is grounded in a transparent, concept-level rationale. The entire model, with only the lightweight prompt modules being trainable, is optimized end-to-end.

### 3.1 FEATURE EXTRACTION AND PROGNOSTIC CONCEPT SELECTION

**Vision Feature Extraction.** Given a radiology image $\boldsymbol{x}_r$ and a pathology whole-slide image (WSI) $\boldsymbol{x}_h$, we first obtain global feature representations using the vision encoders ($V_r$, $V_h$) of powerful, pre-trained medical VLMs. This yields feature vectors $\boldsymbol{f}_r = V_r(\boldsymbol{x}_r) \in \mathbb{R}^{D_{vr}}$ and $\boldsymbol{f}_h = V_h(\boldsymbol{x}_h) \in \mathbb{R}^{D_{vh}}$, where $D_{vr}$ and $D_{vh}$ are the dimensions of the respective vision feature spaces. These encoders remain frozen during training to preserve their rich, pre-trained knowledge.

**Prognostic Concept Selection.** A high-quality set of medical concepts forms the foundation of our interpretable bridge. Given an initial large candidate pool $\mathcal{S}$ (generated via LLMs *per domain*. Details in Appendix C.1), we aim to select a compact subset $\mathcal{C}$ of size $k$ that is both prognostically relevant and semantically diverse. This avoids selecting redundant concepts (e.g., "irregular margins" and "ill-defined borders"). We formalize this as maximizing a submodular objective function (Harshaw et al., 2019; Lin & Bilmes, 2012), which balances relevance $d(\cdot)$ and diversity:

$$F(\mathcal{C}) = \sum_{c \in \mathcal{C}} d(c) + \lambda \sum_{c \in \mathcal{C}} \min_{c' \in \mathcal{C} \setminus \{c\}} (1 - \sigma(\boldsymbol{t}_c, \boldsymbol{t}_{c'})). \tag{1}$$

As this is NP-hard, we approximate the solution with a two-stage greedy algorithm (shown in Fig. 2):

**1. Relevance Ranking:** We first score every candidate concept $c \in \mathcal{S}$ based on its prognostic relevance. We define this relevance score $d(c)$ using the Mutual Information (MI) between the concept's alignment scores and patient labels (Kraskov et al., 2004). For each image $\boldsymbol{x}_i$, an alignment score $a(\boldsymbol{x}_i, c) = (\boldsymbol{t}_c^\top \boldsymbol{f}_i)/(\|\boldsymbol{t}_c\|_2 \|\boldsymbol{f}_i\|_2)$ is computed, where $\boldsymbol{t}_c = T(c)$ is the text embedding of

concept $c$ from the text encoder $T(\cdot)$. We estimate MI via sklearn's kNN-based *mutual_info_classif* (Ross, 2014), which discretizes the continuous alignment internally. We then compute its MI with the label $Y$ as $d(c) = I(\hat{A}_c; Y) = \sum_{y \in Y} \sum_{\hat{a} \in \hat{A}_c} p(\hat{a}, y) \log \frac{p(\hat{a}, y)}{p(\hat{a})p(y)}$. All concepts in $\mathcal{S}$ are then sorted in descending order based on this score.

**2. Diversity Maximization:** We initialize our final concept set $\mathcal{C}$ with the top-ranked concept from the first round. We then iteratively add concepts to $\mathcal{C}$ from the sorted list. At each step, we select the next concept $c^*$ that maximizes semantic diversity with respect to the concepts already chosen, defined as $c^* = \arg\max_{c' \in \mathcal{S} \setminus \mathcal{C}} (\min_{c \in \mathcal{C}} (1 - \sigma(t_{c'}, t_c)))$, where $\sigma(\cdot, \cdot)$ is the cosine similarity between concept text embeddings from the VLM's text encoder $T(\cdot)$. This process continues until $|\mathcal{C}| = k$, yielding the final, high-quality concept sets $\mathcal{C}_{\text{rad}}$ and $\mathcal{C}_{\text{hist}}$.

This selection is performed once with frozen encoders, and tuned prompts (Sec. 3.2) are used only during model training/inference.

## 3.2 CROSS-DOMAIN CONCEPT CO-ADAPTATION

Static concept representations fail to capture how the meaning of a medical finding can shift based on complementary information. To address this, we introduce the **Global-Context-Shared Prompt (GCSP)** strategy, a parameter-efficient tuning method that generates a dynamic, case-specific prefix $\boldsymbol{P}^{\text{tuned}}$ for each concept $c$ before it is processed by the frozen text encoder $T(\cdot)$. Unlike methods that aim to modify static latent features, our approach actively conditions the semantic representation of concepts within each domain, creating a synergistic dialogue *before* fusion. We apply GCSP symmetrically: radiology concepts are conditioned on pathology features and vice versa (Fig. 2). For a radiology (pathology by symmetry) concept, the prefix $\boldsymbol{P}^{\text{tuned}} \in \mathbb{R}^{L \times D_t}$ is a concatenation of three specialized components:

$$\boldsymbol{P}^{\text{tuned}} = \text{Concat}\left(\boldsymbol{P}_G, \boldsymbol{P}_C(\boldsymbol{f}_h), \boldsymbol{P}_S(\boldsymbol{f}_r, \boldsymbol{f}_h)\right). \tag{2}$$

**Global Prompt ($\boldsymbol{P}_G$).** For each concept $c$, we learn a dedicated, domain-specific prompt vector $\boldsymbol{P}_G(c)$. This prompt is shared across all patients within one domain and provides a general adaptation of the concept's pre-trained meaning to the specific nuances of the downstream task.

**Context Prompt ($\boldsymbol{P}_C$).** This prompt is the key to our cross-domain guidance. It allows one modality to influence the interpretation of concepts in the other via a Mixture-of-Experts (MoE) style layer (Shazeer et al., 2017). For a radiology concept $c_r$, the context prompt is generated from the pathology image feature $\boldsymbol{f}_h$. We maintain a learnable pool of $M$ basis prompt vectors $\{\boldsymbol{P}_{C,i}^{\text{basis}}\}_{i=1}^{M}$ shared across all concepts in that domain. A lightweight gating network, $g_r : \mathbb{R}^{D_{vh}} \to \mathbb{R}^{M}$, takes the complementary feature $\boldsymbol{f}_h$ to produce mixture weights: $\boldsymbol{P}_C(\boldsymbol{f}_h) = \sum_{i=1}^{M} \alpha_i \cdot \boldsymbol{P}_{C,i}^{\text{basis}}$, where $\boldsymbol{\alpha} = \text{softmax}(g_r(\boldsymbol{f}_h))$. This mechanism dynamically selects and weights conceptual attributes and provides patient-specific, cross-domain conditioning of concept semantics (see Appendix A.1 for more details). The same mechanism is applied symmetrically for pathology concepts.

**Shared Prompt ($\boldsymbol{P}_S$).** This prompt captures holistic, patient-specific synergy. First, a shared latent vector is produced by a small MLP, $\phi_S$, that takes the concatenated features from both modalities as input: $\boldsymbol{f}_S = \phi_S(\text{Concat}(\boldsymbol{f}_r, \boldsymbol{f}_h))$. This shared feature is then projected by two separate linear layers, to suit different VLMs' context: $\boldsymbol{P}_{S,r}(\boldsymbol{f}_S) = \varphi_{S,r}(\boldsymbol{f}_S)$ and $\boldsymbol{P}_{S,h}(\boldsymbol{f}_S) = \varphi_{S,h}(\boldsymbol{f}_S)$. This provides a unified adjustment signal to all concepts for a given patient, ensuring coherent refinement across domains.

## 3.3 CONCEPT-BASED FUSION AND TASK-SPECIFIC PREDICTION

After generating the composite prompt $\boldsymbol{P}_{\text{tuned}}$ for each concept, we prepend $\boldsymbol{P}_{\text{tuned}}$ to the tokenized concept string. Then we obtain a set of tuned textual embeddings, $\{\tilde{\boldsymbol{t}}_{c_r}\}$ and $\{\tilde{\boldsymbol{t}}_{c_h}\}$, from the text encoder, where $\tilde{\boldsymbol{t}}_c = T(\text{Tuned tokens for } c)$. We then compute two concept score vectors, $\boldsymbol{s}_r \in \mathbb{R}^{|\mathcal{C}_{\text{rad}}|}$ and $\boldsymbol{s}_h \in \mathbb{R}^{|\mathcal{C}_{\text{hist}}|}$, representing the alignment of each image with its corresponding tuned concepts, where the $j$-th element of each vector is computed as: $s_{r,j} = (\boldsymbol{f}_r^\top \tilde{\boldsymbol{t}}_{c_r^{(j)}})/(\|\boldsymbol{f}_r\|_2 \|\tilde{\boldsymbol{t}}_{c_r^{(j)}}\|_2)$ and $s_{h,j} = (\boldsymbol{f}_h^\top \tilde{\boldsymbol{t}}_{c_h^{(j)}})/(\|\boldsymbol{f}_h\|_2 \|\tilde{\boldsymbol{t}}_{c_h^{(j)}}\|_2)$. These score vectors provide an interpretable representation of the patient's condition. The final patient representation $z$ is formed by concatenating the scores from

both domains: $z = \text{Concat}(s_r, s_h)$. This interpretable vector is then passed through a final prediction head, a multi-layer perceptron ($\text{MLP}_{\text{pred}}$), to produce the task-specific output $o = \text{MLP}_{\text{pred}}(z)$. All FM encoders remain frozen and only the prompt modules and prediction head are trainable, totaling 0.5M (0.15% of 307M across both FMs).

**Optimization.** The entire framework is trained end-to-end using a loss function appropriate for the downstream task. For survival analysis, we use the Cox Proportional Hazards model (Cox, 1972) where $o$ is a vector of risk scores. The model is optimized by minimizing the negative log partial likelihood $\mathcal{L}_{\text{cox}} = -\sum_{i:\delta_i=1} \left( r_i - \log \sum_{j \in \mathcal{R}_i} \exp(r_j) \right)$, where $\delta_i = 1$ if the event (e.g., death) was observed and $\delta_i = 0$ if the data is right-censored, $r_i$ is the predicted risk for patient $i$ and $\mathcal{R}_i = \{j | t_j \geq t_i\}$ is the set of patients still at risk at time $t_i$ (Katzman et al., 2018). For a classification task like cancer grading, the prediction head $\text{MLP}_{\text{pred}}$ outputs class logits, and the model is optimized using a standard cross-entropy loss. Note that MI ranking is used only offline for concept selection. At inference, we compute cosine-normalized concept scores with tuned embeddings and no labels or MI are used.

## 4 EXPERIMENTS

We designed a comprehensive set of experiments to validate the core hypotheses of our work. Our evaluation is structured to demonstrate that **(1)** our concept-based fusion paradigm, CTF, surpasses state-of-the-art methods that rely on static or independently adapted latent features; **(2)** the performance gains are primarily driven by our novel GCSP strategy, which enables a dynamic, cross-domain dialogue; and **(3)** CTF yields predictions that are not only accurate but also interpretable, robust across diverse clinical tasks, and grounded in clinically plausible reasoning.

### 4.1 EXPERIMENTAL SETUP

**Datasets.** We evaluate CTF on two distinct and clinically vital predictive tasks: survival analysis and cancer grading. We curated four datasets spanning different cancer types and imaging modalities. For survival analysis, we use three cohorts with paired imaging and clinical data: TCGA-LGG[2], TCGA-GBM, and private Center1-GC (Gastric Cancer). For cancer grading, we evaluate performance on three cohorts, including private Center2-CHS (Chondrosarcoma). The specific classifications include 3-way WHO grades for the brain tumor cohorts (LGG and GBM merged) and 5-way TNM T-stage for the gastric cancer cohort, a particularly challenging benchmark task. We report mean±sd over 10 stratified train/val/test splits. Besides, we perform paired t-tests between CTF and the strongest baseline per task in Appendix D.4.

**Implementation Details.** To instantiate our framework, we selected *domain-expert foundation models* to maximize clinical relevance: BiomedCLIP (Zhang et al., 2023) for radiology and CONCH (Lu et al., 2024) for pathology. By keeping their vision encoders frozen, we leverage their specialized knowledge bases and focus our method on creating a synergistic dialogue between them. For each domain, we selected $k = 256$ concepts using the designed strategy (Sec. 3.1) and inserted $L = 12$ tunable tokens (Sec. 3.2). The whole framework, including all prompt generators, is trained end-to-end. Full implementation, hyperparameter details, and hardware/GPU usage are in Appendix A.2 and Appendix A.3.

Table 1: Survival prediction performance (C-index ↑) on three datasets. Best performance is in **bold**, second-best is underlined.

| Model | TCGA-LGG | TCGA-GBM | Center1-GC |
|---|---|---|---|
| *Unimodal Baselines* | | | |
| Radiology-Only | $0.598 \pm 0.128$ | $0.477 \pm 0.055$ | $0.614 \pm 0.052$ |
| ABMIL (Ilse et al., 2018) | $0.669 \pm 0.101$ | $0.480 \pm 0.093$ | $0.590 \pm 0.030$ |
| CLAM (Lu et al., 2021) | $0.689 \pm 0.108$ | $0.497 \pm 0.068$ | $0.631 \pm 0.060$ |
| TransMIL (Shao et al., 2021) | $0.682 \pm 0.121$ | $0.503 \pm 0.055$ | $0.613 \pm 0.066$ |
| ACMIL (Zhang et al., 2024b) | $0.678 \pm 0.142$ | $0.519 \pm 0.057$ | $0.628 \pm 0.083$ |
| *Multimodal Latent Fusion Baselines* | | | |
| Concat-Fusion | $0.674 \pm 0.112$ | $0.515 \pm 0.070$ | $0.626 \pm 0.048$ |
| Cross-Attention | $0.685 \pm 0.108$ | $0.527 \pm 0.068$ | $0.631 \pm 0.060$ |
| MOTCAT (Xu & Chen, 2023) | $0.571 \pm 0.080$ | $\underline{0.563 \pm 0.108}$ | $0.622 \pm 0.040$ |
| PIBD (Zhang et al., 2024a) | $0.687 \pm 0.123$ | $0.531 \pm 0.061$ | $0.638 \pm 0.058$ |
| *Multimodal Adaptive Fusion Baseline* | | | |
| M4Survive (Lee et al., 2025) | $\underline{0.709 \pm 0.112}$ | $0.545 \pm 0.072$ | $\underline{0.642 \pm 0.065}$ |
| **CTF (Ours)** | $\mathbf{0.713 \pm 0.103}$ | $\mathbf{0.579 \pm 0.063}$ | $\mathbf{0.665 \pm 0.061}$ |

---

[2]https://www.cancer.gov/tcga/

Table 2: Cancer Grading Performance (AUC↑ and ACC↑). Best performance is in **bold**, second-best is underlined.

| Model | TCGA-GBMLGG (3-way) | | Center2-CHS (5-way) | | Center1-GC (5-way) | |
|---|---|---|---|---|---|---|
| | AUC↑ | ACC↑ | AUC↑ | ACC↑ | AUC↑ | ACC↑ |
| *Unimodal Baselines* | | | | | | |
| Radiology-Only | $0.776 \pm 0.059$ | $0.624 \pm 0.064$ | $0.679 \pm 0.069$ | $0.429 \pm 0.091$ | $0.595 \pm 0.087$ | $0.341 \pm 0.080$ |
| ABMIL (Ilse et al., 2018) | $0.855 \pm 0.050$ | $0.667 \pm 0.076$ | $0.770 \pm 0.092$ | $0.493 \pm 0.138$ | $0.609 \pm 0.063$ | $0.384 \pm 0.053$ |
| CLAM (Lu et al., 2021) | $0.860 \pm 0.048$ | $0.681 \pm 0.070$ | $0.775 \pm 0.089$ | $0.512 \pm 0.130$ | $0.628 \pm 0.055$ | $0.390 \pm 0.051$ |
| TransMIL (Shao et al., 2021) | $0.864 \pm 0.050$ | $0.684 \pm 0.068$ | $0.781 \pm 0.085$ | $0.518 \pm 0.128$ | $0.625 \pm 0.058$ | $0.388 \pm 0.049$ |
| ACMIL (Zhang et al., 2024b) | $0.853 \pm 0.046$ | $0.680 \pm 0.069$ | $0.779 \pm 0.046$ | $0.515 \pm 0.129$ | $0.619 \pm 0.062$ | $0.389 \pm 0.054$ |
| *Multimodal Latent Fusion Baselines* | | | | | | |
| Concat-Fusion | $0.858 \pm 0.038$ | $0.687 \pm 0.062$ | $0.805 \pm 0.075$ | $0.535 \pm 0.115$ | $0.629 \pm 0.051$ | $0.391 \pm 0.048$ |
| Cross-Attention | $\underline{0.868 \pm 0.030}$ | $\underline{0.695 \pm 0.059}$ | $0.817 \pm 0.071$ | $0.581 \pm 0.110$ | $0.635 \pm 0.048$ | $\underline{0.394 \pm 0.049}$ |
| MOTCAT (Xu & Chen, 2023) | $0.865 \pm 0.025$ | $0.657 \pm 0.053$ | $0.826 \pm 0.078$ | $0.612 \pm 0.120$ | $0.641 \pm 0.050$ | $0.390 \pm 0.052$ |
| *Multimodal Adaptive Fusion Baseline* | | | | | | |
| M4Survive (Lee et al., 2025) | $0.861 \pm 0.031$ | $0.691 \pm 0.061$ | $\underline{0.830 \pm 0.075}$ | $\underline{0.626 \pm 0.115}$ | $\underline{0.649 \pm 0.052}$ | $0.390 \pm 0.051$ |
| CTF (Ours) | $\mathbf{0.903 \pm 0.028}$ | $\mathbf{0.718 \pm 0.063}$ | $\mathbf{0.854 \pm 0.081}$ | $\mathbf{0.698 \pm 0.164}$ | $\mathbf{0.660 \pm 0.049}$ | $\mathbf{0.401 \pm 0.057}$ |

## 4.2 BASELINES

We compared CTF against a comprehensive set of models. Unimodal baselines include several high-performing single-modality methods (ABMIL (Ilse et al., 2018), CLAM (Lu et al., 2021), etc.). Fusion baselines include: (1) Simple Fusion methods (Concat-Fusion, Cross-Attention); (2) State-of-the-Art (SOTA) Latent Fusion methods that fuse static features (MOTCAT (Xu & Chen, 2023), PIBD (Zhang et al., 2024a)); and (3) a SOTA Adaptive Fusion method that fine-tunes model weights (M4Survive (Lee et al., 2025)). Detailed descriptions of each baseline are in Appendix A.3. For fairness, all baselines share the same frozen vision encoders and survival/classification heads, and only fusion modules differ. Adaptation specifics for MOTCAT/PIBD are in Appendix A.4.

## 4.3 QUANTITATIVE RESULTS

**Superior Survival Prediction.** As presented in Table 1, CTF consistently achieves state-of-the-art performance, outperforming all baselines in our study on the three survival prediction cohorts. Notably, it achieves a C-index (Harrell et al., 1982) of 0.713 on TCGA-LGG, surpassing the strongest adaptive baseline, M4Survive, and the strongest latent fusion baseline, PIBD, by 3.8%. The results reveal a clear hierarchy of fusion strategies. While all multimodal methods generally outperform unimodal approaches, affirming the value of data integration, the key distinction lies in how fusion is performed. Advanced latent fusion models like PIBD show respectable gains over simple concatenation but are ultimately limited by their reliance on static, pre-extracted features. Adaptive methods like M4Survive improve upon this by fine-tuning architectural components.

However, CTF's superior performance suggests a fundamental advantage. Instead of simply fusing latent vectors or adapting architectural blocks, CTF performs *semantic co-adaptation*. By dynamically tuning the meaning of clinical concepts in one domain based on context from the other before fusion, it achieves a deeper, more synergistic integration. This consistent performance gain across diverse cancer types aligns with our hypothesis: dynamic, concept-based co-adaptation is a more effective paradigm for multimodal fusion than static latent fusion or independent architectural adaptation.

**Generalization to Cancer Grading.** As shown in Table 2, CTF again achieves state-of-the-art performance, outperforming all baselines across the three datasets. On average, CTF obtains an AUC improvement of 3.6% over the strongest fusion baseline (MOTCAT). This is a significant result, as cancer grading relies on identifying distinct morphological and cellular patterns. CTF's success suggests its ability to learn a rich, concept-based dialogue between radiology's macro-structural views and pathology's micro-cellular details is highly effective for this classification task. This robust performance validates the broader utility of our concept-tuning and fusion paradigm.

Table 3: Ablation on the Center1-GC dataset for survival prediction and tumor grading. Metrics are reported as mean with standard deviation in parentheses. Best numbers are in bold. $\Delta$ denotes absolute change vs. the full CTF model.

| Category | Variant | Survival Prediction | | Tumor Grading | |
|---|---|---|---|---|---|
| | | C-index (↑) | $\Delta$ | AUC (↑) | $\Delta$ |
| Reference | CTF (Full Model) | **0.665 (0.061)** | (-) | **0.660 (0.049)** | (-) |
| Prompt Components | w/o Context Prompt ($P_C$) | 0.629 (0.058) | (-0.036) | 0.635 (0.047) | (-0.025) |
| | w/o Shared Prompt ($P_S$) | 0.653 (0.063) | (-0.012) | 0.651 (0.056) | (-0.009) |
| | w/o Global Prompt ($P_G$) | 0.642 (0.051) | (-0.023) | 0.640 (0.048) | (-0.020) |
| Tuning Strategy | Static Concepts (CBM) | 0.586 (0.049) | (-0.079) | 0.622 (0.055) | (-0.038) |
| | Static Concepts + Prompt Tuning | 0.638 (0.059) | (-0.027) | 0.635 (0.052) | (-0.025) |
| Concept Selection | Random Selection | 0.622 (0.053) | (-0.043) | 0.654 (0.053) | (-0.006) |
| | Top-MI Selection (relevance-only) | 0.646 (0.060) | (-0.019) | 0.642 (0.049) | (-0.018) |
| Backbone Sensitivity | CTF (General, CLIP + CLIP) | 0.621 (0.055) | (-0.044) | 0.615 (0.054) | (-0.045) |
| | CTF (Hybrid, CLIP + CONCH) | 0.639 (0.059) | (-0.026) | 0.643 (0.050) | (-0.017) |
| | CTF (Expert, BiomedCLIP + MUSK) | 0.680 (0.064) | (+0.015) | 0.658 (0.053) | (-0.002) |
| | CTF (Expert, BiomedCLIP + PLIP) | 0.627 (0.063) | (-0.038) | 0.636 (0.052) | (-0.029) |

## 4.4 ANALYSIS AND ABLATION STUDIES

**The Critical Role of Cross-Domain Dialogue.** As shown in Table 3, the most impactful ablation was the removal of the Context Prompt ($P_C$), which embodies our cross-domain guidance mechanism. This single change caused the most significant performance degradation, with the C-index plummeting by 0.036. This result provides direct empirical evidence that forcing each modality to be "aware" of the other during concept interpretation is the primary driver of CTF's success. The removal of the Global (task-specific) and Shared (synergistic) prompts also led to performance drops, confirming that all components of the GCSP strategy contribute meaningfully to the performance.

**Dynamic Tuning vs. Static Concepts.** We then tested a model using "Static Concepts" without any prompt tuning, which is analogous to a standard Concept Bottleneck Model (CBM) (Koh et al., 2020). Specifically, the "static concepts (CBM)" variant uses fixed, pre-trained concepts and feeds their scores directly to an MLP head for downstream predictions. As in Table 3, we found the model performed poorly (C-index 0.586), demonstrating that simply using concepts as an intermediate layer is insufficient. The dynamic adaptation enabled by GCSP is paramount. Interestingly, this aligns with recent findings that show modern, flexible prototype-based methods are moving beyond the rigid CBM structure (Chen et al., 2025b). Our work contributes a novel cross-modal tuning mechanism to this emerging paradigm.

**Concept Selection and Backbone Choice.** The inputs to our framework are also critical. We compared our prognostic and diversity-aware concept selection against two alternatives: Random Selection and Top-MI Selection (relevance-only). As shown in Table 3, our principled strategy significantly outperforms both, validating the need to select concepts that are both prognostically relevant and semantically diverse. Furthermore, to confirm the value of domain-specific foundation models, we replaced the expert BiomedCLIP and CONCH encoders with general-purpose CLIP models. This "Generalist" setup led to a substantial performance drop (C-index from 0.665 to 0.621), confirming that CTF's ability to bridge modalities is maximized when it operates on the

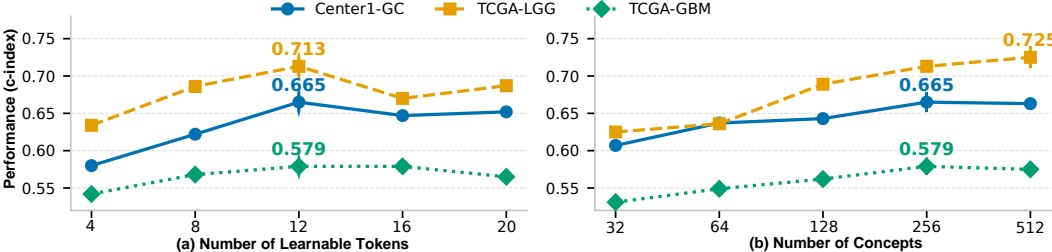

Figure 3: Hyperparameter sensitivity analysis on all three datasets for survival prediction. (a) Performance (C-index) versus the number of learnable tokens ($L$) per prompt component. (b) Performance versus the number of selected concepts ($k$) per domain.

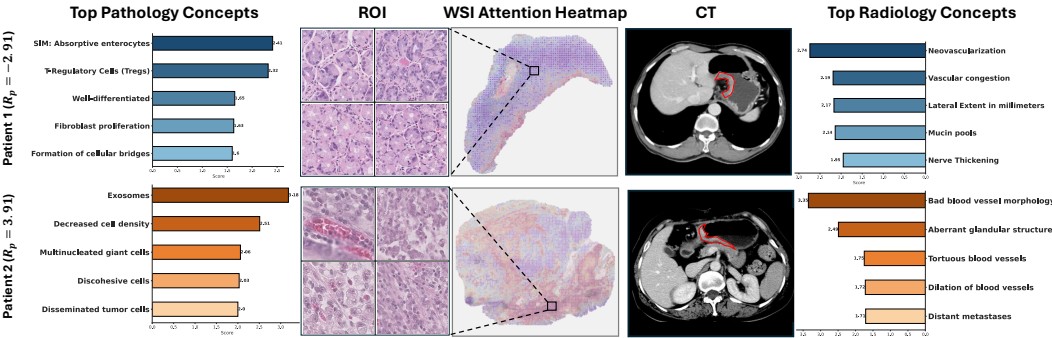

Figure 4: Concept-based Interpretation of CTF Predictions. Visualization of top 5 pathology and radiology concept scores for (top) a low-risk patient censored at a late time point and (bottom) a high-risk patient with an early event from the Center1-GC dataset.

rich representations of expert models. When fixing radiology to CLIP and varying pathology VLMs ('Hybrid' rows in Table 3), we observe that pathology-specific VLMs (CONCH, MUSK (Xiang et al., 2025), PLIP (Huang et al., 2023)) outperform generic CLIP, with MUSK slightly ahead of CONCH on the survival task of Center1-GC.

**Parameter Sensitivity.** We analyzed CTF's sensitivity to two key hyperparameters in our GCSP strategy: the prompt length $L$ and the number of concepts $k$. As shown in Figure 3, we evaluated performance across all three survival cohorts. In Figure 3a, we varied the length of the tunable prompt from 4 to 20. Performance is generally stable, with a slight peak at $L = 12$. Shorter prompts may lack expressive power, while longer ones increase parameter counts without a clear benefit, validating our choice of $L = 12$ as an efficient and effective setting. In Figure 3b, we varied the number of selected concepts per modality from 32 to 512. Performance peaks at $k = 256$. Too few concepts (e.g., $k = 64$) fail to capture sufficient prognostic information, while too many ($k = 512$) may introduce noise and a larger computation cost without improving, and in some cases slightly degrading, performance. This confirms $k = 256$ might be an optimal choice.

## 4.5 QUALITATIVE ANALYSIS AND INTERPRETABILITY

Beyond quantitative benchmarks, a critical goal of CTF is to provide transparent and trustworthy predictions. We conducted a series of qualitative analyses to demonstrate that our model's reasoning is grounded in clinically relevant patterns.

**Patient-Level Rationale.** Because $z$ is a vector of concept scores, we can inspect which radiology and pathology concepts dominate the prediction for a given patient. Figure 4 visualizes final concept scores for two Center1-GC patients: one low-risk censored late, and one high-risk with an early event. The low-risk patient (Figure 4 top) suggests a less aggressive phenotype, such as "Well-differentiated tumor". In contrast, the high-risk patient (Figure 4 bottom) shows high scores for aggressive concepts like "Disseminated tumor cells" and "Bad blood vessel morphology", providing a clear and alarming prognostic signal. This demonstrates CTF provides accurate predictions with a transparent, clinically relevant rationale based on meaningful concepts.

**Semantic Drift.** We compared the concept embedding before and after tuning. To quantify how GCSP modifies concept semantics, we compute, for each concept $c$, the cosine similarity $s_c = \cos(\tilde{\mathbf{t}}_c, \mathbf{t}_c)$ between the tuned embedding $\tilde{\mathbf{t}}_c$ and its original text embedding $\mathbf{t}_c$ from the frozen encoder. For Center1-GC's radiology concepts, the distribution of $s_c$ concentrates well above $0.5$ with a median of $0.633$ (Fig. 5a), indicating moderate, task-sharpened shifts rather than wholesale redefinition. The t-SNE visualization (Fig. 5b) corroborates this pattern, showing coherent displacement without mode collapse. Practically, this bounded drift preserves the interpretability of the original human-readable concepts while enabling measurable gains in downstream prediction.

**Visualizing Cross-Modal Influence.** We assess whether GCSP captures clinically meaningful dependencies by grouping selected concepts into high-level radiology and pathology categories and

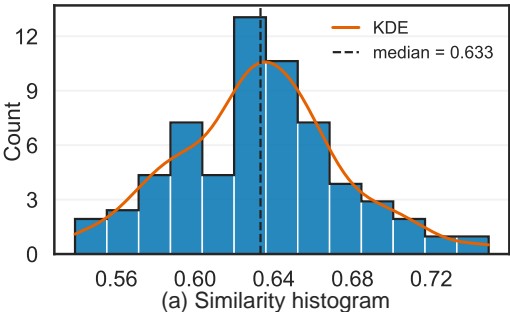 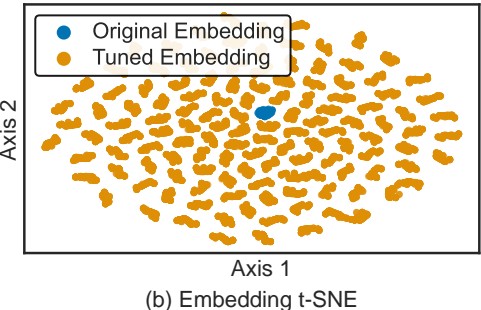

Figure 5: Semantic drift of concept embeddings. (a) Histogram of cosine similarity between tuned and original concept embeddings (median 0.633) indicates task-sharpened but semantically consistent shifts. (b) t-SNE shows systematic displacement without mode collapse.

quantifying how the radiology-to-pathology context prompt $P_C$ alters pathology concept scores (vs. a zeroed $P_C$).

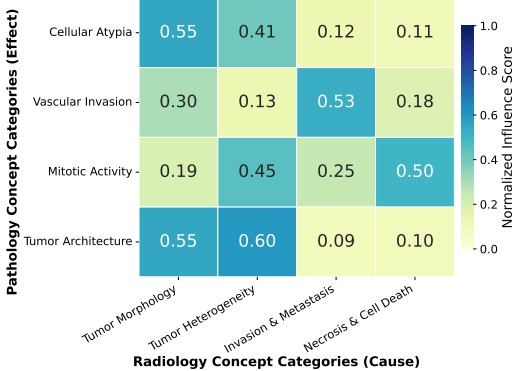

Figure 6: Cross-modal influence analysis on the Center1-GC dataset. The heatmap shows the changes in normalized influence score, representing how much the presence of a radiology concept category (columns) affects the scores of pathology concept categories (rows) via ($P_C$).

The Fig. 6 heatmap shows clinically plausible relationships learned by the model. For instance, radiology concepts related to "Tumor Morphology" and "Invasion & Metastasis" strongly amplify pathology concepts of "Cellular Atypia" and "Vascular Invasion," mimicking a pathologist's reasoning process where macroscopic signs of aggression prompt a closer search for microscopic evidence (Tomaszewski & Gillies, 2021). Besides, radiology's "Necrosis & Cell Death" has minimal influence on pathology's "Mitotic Activity". This is clinically sound: while large-scale necrosis is visible on a CT scan, it is a poor predictor of the specific rate of cell division (mitotic count), which is only assessable under a microscope (Bosman et al., 2010).

## 5  CONCLUSION AND LIMITATIONS

We presented CTF, a parameter-efficient, concept-based multimodal co-adaptation framework that bridges radiology and pathology FMs. Via the Global–Context–Shared Prompt (GCSP), CTF dynamically tunes clinically grounded concepts with task, cross-domain, and shared patient context, aligning representations before fusion and yielding transparent, concept-level predictions. Ablations confirm the primacy of cross-domain context and principled concept selection, and qualitative analyses reveal clinically plausible rationales and influence patterns.

Limitations include reliance on a predefined concept pool and paired data. Furthermore, the modest absolute performance on the difficult 5-way gastric cancer grading task underscores that it is not ready for clinical deployment in this specific scenario, serving as a benchmark for methodological comparison. Our current concept pool is generated once by an LLM and then fixed throughout training. While GCSP dynamically tunes concept semantics, it does not introduce new labels or expand the concept vocabulary. Beyond tuning semantics, future work will (i) periodically refresh/expand the concept pool using model-derived importance and LLM-decoding of tuned embeddings, and (ii) handle missing modalities via partial-paired training and concept imputation.

## ACKNOWLEDGMENT

This work was supported in part by the Research Grants Council of Hong Kong (27206123, 17200125, C5055-24G, and T45-401/22-N), the Hong Kong Innovation and Technology Fund (GHP/318/22GD), Guangdong Natural Science Fund (No. 2024A1515011875), the Start-up Fund of The Hong Kong Polytechnic University (No. P0045999), the Seed Fund of the Research Institute for Smart Ageing (No. P0050946), and Tsinghua-PolyU Joint Research Initiative Fund (No. P0056509), and PolyU UGC funding (No. P0053716).

## ETHICS STATEMENT

This research utilized both public and private medical datasets. All data from The Cancer Genome Atlas (TCGA) are publicly available and de-identified. The in-house datasets (Center1-GC and Center2-CHS) were collected under protocols approved by the local Institutional Review Board (IRB), with all patient data fully anonymized before use in this study. Informed consent was obtained from all participants. Our study strictly adheres to data privacy and protection regulations, as detailed in Appendix B.

## REPRODUCIBILITY STATEMENT

To ensure the reproducibility of our results, we have made our code anonymously available at: `https://anonymous.4open.science/r/CTF-27C2`. The main paper provides a detailed description of our methodology in Section 3. The appendix further provides pseudocode for our core algorithms, comprehensive architectural and training details, including hyperparameters, and specifics on baseline implementations (Appendix A).

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

# Appendix

## A    FRAMEWORK AND IMPLEMENTATION DETAILS

This section provides a detailed breakdown of our framework's implementation, including pseudocode for core algorithms, expanded architectural descriptions, and specifics on baseline implementations to ensure full reproducibility.

### A.1    ALGORITHM PSEUDOCODE

To further clarify the mechanics of our proposed method, we provide pseudocode for the two main stages of the CTF framework: the greedy concept selection process (Algorithm 1), the end-to-end forward pass for a single patient (Algorithm 2), and the MoE layer to generate context prompts.

---

**Algorithm 1** Greedy Algorithm for Prognostic and Diverse Concept Selection

---

1: **Input:** Initial concept pool $\mathcal{S}$, target concept set size $k$, training data images $\{\boldsymbol{x}_i\}$, survival outcomes $\{y_i, \delta_i\}$.
2: **Input:** Frozen vision encoder $V(\cdot)$ and text encoder $T(\cdot)$.
3: **Initialize:** Final concept set $\mathcal{C} \leftarrow \emptyset$.
4: **Initialize:** Relevance scores $D \leftarrow \{\}$.
5: Pre-compute image features $\boldsymbol{f}_i = V(\boldsymbol{x}_i)$ and concept embeddings $\boldsymbol{t}_c = T(c)$ for all $i, c$.
6: {— Stage 1: Relevance Ranking —}
7: **for** each concept $c \in \mathcal{S}$ **do**
8:     Compute alignment scores $A_c = \{a(\boldsymbol{x}_i, c) = \boldsymbol{t}_c^\top \boldsymbol{f}_i\}_{i=1}^N$.
9:     Discretize scores $A_c$ to get $\hat{A}_c$. Binarize survival outcomes to get $Y_{\text{bin}}$.
10:     Compute mutual information $d(c) = I(\hat{A}_c; Y_{\text{bin}})$.
11:     Add $(c, d(c))$ to $D$.
12: **end for**
13: Sort concepts in $\mathcal{S}$ descending by their relevance score $d(c)$. Let the sorted list be $\mathcal{S}_{\text{sorted}}$.
14: {— Stage 2: Diversity Maximization —}
15: Add the top-ranked concept from $\mathcal{S}_{\text{sorted}}$ to $\mathcal{C}$.
16: **while** $|\mathcal{C}| < k$ **do**
17:     **Initialize:** "max_diversity" $\leftarrow -1$, "next_concept" $\leftarrow$ null.
18:     **for** each concept $c' \in \mathcal{S}_{\text{sorted}} \setminus \mathcal{C}$ **do**
19:         Compute max similarity to current set: $\max\_sim = \max_{c \in \mathcal{C}} \sigma(\boldsymbol{t}_{c'}, \boldsymbol{t}_c)$.
20:         Diversity score = $1 - \max\_sim$.
21:         **if** Diversity score > "max_diversity" **then**
22:             "max_diversity" $\leftarrow$ Diversity score.
23:             "next_concept" $\leftarrow c'$.
24:         **end if**
25:     **end for**
26:     Add "next_concept" to $\mathcal{C}$ and remove it from $\mathcal{S}_{\text{sorted}}$.
27: **end while**
28: **Output:** Final selected concept set $\mathcal{C}$.

---

### A.2    ARCHITECTURAL AND TRAINING DETAILS

To ensure full reproducibility, we provide an expanded set of implementation details. All experiments were conducted on an HPC server equipped with NVIDIA L40s GPUs (48GB VRAM). See Table 5 for detailed hyperparameters.

**Image Preprocessing.**

- **Pathology (WSIs):** Whole-slide images were processed at 20x magnification. We segmented the tissue foreground via Otsu's thresholding. The WSI was then tiled into non-overlapping $256 \times 256$ patches. Following the CLAM methodology (Lu et al., 2021), we used contour filtering to remove

---

**Algorithm 2** CTF Forward Pass for a Single Patient

---

1: **Input:** Radiology image $\boldsymbol{x}_r$, pathology image $\boldsymbol{x}_h$.
2: **Input:** Selected radiology concepts $\mathcal{C}_{\text{rad}}$, pathology concepts $\mathcal{C}_{\text{hist}}$.
3: **Models:** Frozen encoders $V_r, V_h, T_r, T_h$; Learnable modules: $\boldsymbol{P}_G, \boldsymbol{P}_C^{\text{basis}}, g_r, g_h, \phi_S, \varphi_{S,r}, \varphi_{S,h}, \text{MLP}_{\text{pred}}$.
4: {— 1. Feature Extraction —}
5: $\boldsymbol{f}_r \leftarrow V_r(\boldsymbol{x}_r), \boldsymbol{f}_h \leftarrow V_h(\boldsymbol{x}_h)$.
6: {— 2. GCSP and Concept Tuning (Example for Radiology) —}
7: **for** each concept $c_r \in \mathcal{C}_{\text{rad}}$ **do**
8:     **Global Prompt:** $\boldsymbol{P}_{G_r} \leftarrow$ Look up learnable prompt for $c_r$.
9:     **Shared Prompt:** $\boldsymbol{f}_S \leftarrow \phi_S(\text{Concat}(\boldsymbol{f}_r, \boldsymbol{f}_h)), \boldsymbol{P}_{S,r} \leftarrow \varphi_{S,r}(\boldsymbol{f}_S)$.
10:     **Context Prompt:** $\boldsymbol{\alpha} \leftarrow \text{softmax}(g_r(\boldsymbol{f}_h)), \boldsymbol{P}_{C_r} \leftarrow \sum_i \alpha_i \cdot \boldsymbol{P}_{C,r,i}^{\text{basis}}$.
11:     **Combine:** $\boldsymbol{P}_r^{\text{tuned}} \leftarrow \text{Concat}(\boldsymbol{P}_{G_r}, \boldsymbol{P}_{C_r}, \boldsymbol{P}_{S,r})$.
12:     **Tune:** Get original tokens for $c_r$. Prepend $\boldsymbol{P}_r^{\text{tuned}}$.
13:     $\tilde{\boldsymbol{t}}_{c_r} \leftarrow T_r(\text{Tuned concept tokens for } c_r)$.
14: **end for**
15: *Repeat symmetrically for each concept $c_h \in \mathcal{C}_{hist}$ to get $\tilde{\boldsymbol{t}}_{c_h}$.*
16: {— 3. Concept Scoring and Fusion —}
17: Compute radiology scores: $s_{r,j} = \text{cosine\_sim}(\boldsymbol{f}_r, \tilde{\boldsymbol{t}}_{c_{r_j}})$. Let $\boldsymbol{s}_r = [s_{r,1}, ..., s_{r,k}]$.
18: Compute pathology scores: $s_{h,j} = \text{cosine\_sim}(\boldsymbol{f}_h, \tilde{\boldsymbol{t}}_{c_{h_j}})$. Let $\boldsymbol{s}_h = [s_{h,1}, ..., s_{h,k}]$.
19: $\boldsymbol{z} \leftarrow \text{Concat}(\boldsymbol{s}_r, \boldsymbol{s}_h)$.
20: {— 4. Prediction —}
21: $\boldsymbol{o} \leftarrow \text{MLP}_{\text{pred}}(\boldsymbol{z})$.
22: **Output:** Task-specific output $\boldsymbol{o}$ (e.g., risk score for survival analysis).

---

background/whitespace patches before feature extraction with the CONCH vision encoder. A small attention pooling head is learned for feature aggregation.

- **Radiology (CT/MRI):** For the in-house Center1-GC and Center2-CHS datasets, tumor regions on CT/MRI scans were manually segmented by an expert radiologist. For the public TCGA datasets, we used the entire axial slice containing the largest tumor cross-section, as brain tumors typically occupy a large and central portion of the image. All radiology images were resized to 224×224, normalized to [0, 1], and then normalized using ImageNet statistics before being fed to the BiomedCLIP vision encoder.

**Compute and memory footprint.** We report peak GPU memory and full training time (batch size 1). $L_{\text{seq}}$ denotes the sequence length (all tokens) and $k_{\text{rad}}, k_{\text{path}}$ the number of concepts per domain.

Table 4: GPU memory usage and wall-clock time per epoch for CTF under different configurations. All results are measured with batch size 1. $L_{\text{seq}}$ denotes the sequence length (tokens) and $k_{\text{rad}}, k_{\text{path}}$ denote the number of concepts for radiology and pathology, respectively.

| Configuration | GPU | $(L_{\text{seq}}, k_{\text{rad}}, k_{\text{path}})$ | Peak memory (GB) | Time (hour) | C-index |
|---|---|---|---|---|---|
| Default (recommended) | NVIDIA L40s 48GB | (128, 256, 256) | 28.7 | 18.9 | 0.665 ± 0.061 |
| 3090-friendly (A) | RTX 3090 24GB | (64, 256, 256) | 19.2 | 11.6 | 0.642 ± 0.053 |
| 3090-friendly (B) | RTX 3090 24GB | (128, 128, 256) | 22.0 | 15.5 | 0.656 ± 0.066 |

## A.3 BASELINE IMPLEMENTATION DETAILS

For a fair comparison, all baselines were implemented using the same frozen vision encoders (BiomedCLIP, CONCH) as our CTF model to extract initial features.

- **Unimodal Models:** For pathology-based unimodal models (ABMIL, CLAM, TransMIL, ACMIL), we used the official publicly available codebases and adapted them to the survival prediction task using a Cox loss final layer. For the *Radiology-Only* baseline, the single feature vector was passed through an MLP identical to our prediction head.

---

**Algorithm 3** Noisy Top-$k$ MoE for Context Prompt Generation

---

1: **Input:** Complementary-modality feature for a mini-batch $X \in \mathbb{R}^{B \times D}$; basis prompts $\{P_i\}_{i=1}^{M}$, $P_i \in \mathbb{R}^{L \times D_t}$; gating parameters $W_{\text{gate}}, W_{\text{noise}} \in \mathbb{R}^{D \times M}$; top-$k$ value $k$ ($k \leq M$); noise $\epsilon > 0$; training flag train; loss coefficient $\lambda$.
2: Compute clean logits: Clean $\leftarrow XW_{\text{gate}}$ {shape $(B \times M)$}
3: **if** train and noisy gating enabled **then**
4:    Std $\leftarrow$ softplus$(XW_{\text{noise}}) + \epsilon$ {$(B \times M)$}
5:    Sample noise: $\Delta \sim \mathcal{N}(0, \text{Std}^2)$
6:    Logits $\leftarrow$ Clean $+ \Delta$
7: **else**
8:    Logits $\leftarrow$ Clean
9: **end if**
10: Convert to probabilities: $P \leftarrow$ softmax(Logits, dim $= 1$) {$(B \times M)$}
11: Initialize $G \leftarrow \mathbf{0}_{B \times M}$
12: **for** $b = 1$ to $B$ **do**
13:    Select top-$k$ indices and values from $P[b, :]$: $(\mathcal{S}_b, v_b) \leftarrow$ TopK$(P[b, :], k)$
14:    Normalize within top-$k$: $\alpha_b \leftarrow v_b/(\sum v_b + 10^{-6})$ {$\alpha_b \in \mathbb{R}^k$}
15:    Set sparse gates: $G[b, \mathcal{S}_b] \leftarrow \alpha_b$
16:    Compute context prompt: $P_C^{(b)} \leftarrow \sum_{i \in \mathcal{S}_b} \alpha_{b,i} P_i$ {$P_C^{(b)} \in \mathbb{R}^{L \times D_t}$}
17: **end for**
18: Importance per expert: Imp $\leftarrow \sum_{b=1}^{B} G[b, :]$ {$(M)$}
19: **if** train and noisy gating enabled and $k < M$ **then**
20:    Compute soft load (expected assignment count) via NoisyTopK (see Alg. 4): Load $\leftarrow \sum_{b=1}^{B}$ ProbInTopK$(\text{Clean}[b, :], \text{Logits}[b, :], \text{Std}[b, :], P[b, \mathcal{S}_b])$ {$(M)$}
21: **else**
22:    Hard load: Load $\leftarrow \sum_{b=1}^{B} \mathbb{I}[G[b, :] > 0]$ {$(M)$}
23: **end if**
24: Coefficient of variation squared: $\text{CV}^2(z) \leftarrow \text{Var}(z)/(\text{Mean}(z)^2 + 10^{-10})$
25: Auxiliary loss: $\mathcal{L}_{\text{moe}} \leftarrow \lambda(\text{CV}^2(\text{Imp}) + \text{CV}^2(\text{Load}))$
26: **Output:** Sparse gates $G \in \mathbb{R}^{B \times M}$, per-sample context prompts $\{P_C^{(b)}\}_{b=1}^{B}$, auxiliary load-balancing loss $\mathcal{L}_{\text{moe}}$.

---

**Algorithm 4** ProbInTopK for Noisy Gating (expected soft load)

---

1: **Input:** Clean logits $c \in \mathbb{R}^M$, noisy logits $n \in \mathbb{R}^M$, noise std $\sigma \in \mathbb{R}^M$ (all for one sample), top-$k$ values topv $\in \mathbb{R}^{k+1}$ from $n$ (descending).
2: Let $\tau_{\text{in}} \leftarrow$ topv$[k]$ and $\tau_{\text{out}} \leftarrow$ topv$[k-1]$ {thresholds for in/out}
3: **for** $j = 1$ to $M$ **do**
4:    **if** $n[j] > \tau_{\text{in}}$ **then**
5:       prob$[j] \leftarrow \Phi((c[j] - \tau_{\text{in}})/\sigma[j])$
6:    **else**
7:       prob$[j] \leftarrow \Phi((c[j] - \tau_{\text{out}})/\sigma[j])$
8:    **end if**
9: **end for**
10: **Output:** Vector prob $\in \mathbb{R}^M$ with prob$[j] = \mathbb{P}(j \in \text{Top-}k)$.

---

- **Simple Latent Fusion:** *Concat-Fusion* involved concatenating $\boldsymbol{f}_r$ and $\boldsymbol{f}_h$ and feeding them to the prediction head. *Cross-Attention* used a standard transformer encoder layer where features from one modality formed the query and features from the other formed the key/value, followed by concatenation of the attended features.

- **SOTA Latent Fusion:** For *MOTCAT* and *PIBD*, we re-implemented the core fusion mechanisms described in their respective papers, placing them between our frozen feature extractors and the final prediction head. We performed a hyperparameter search for key parameters, such as the number of attention heads for MOTCAT and the $\beta$ coefficient for PIBD's bottleneck.

Table 5: Hyperparameters for the CTF model components.

| Parameter | Value | Description |
|---|---|---|
| Optimizer | AdamW | - |
| Learning Rate | 1e-3 | Initial learning rate. |
| Weight Decay | 1e-6 | - |
| Batch Size | 1 | - |
| Epochs | 100 | With early stopping (patience=6 on val-loss). |
| Tunable Prompt Length ($L$) | 12 | For all prompt types (Global, Context, Shared). |
| Context Prompt Pool Size ($M$) | 16 | Number of basis prompt vectors. |
| Gating Network ($g_r, g_h$) | 2-layer MLP ($512 \rightarrow 128 \rightarrow 16$) | With ReLU activation. |
| Shared Prompt Generator ($\phi_S$) | 2-layer MLP ($1024 \rightarrow 256 \rightarrow 128$) | With ReLU activation. |
| Prediction Head (MLP$_{\text{pred}}$) | 2-layer MLP ($512 \rightarrow 128 \rightarrow$ #classes) | With ReLU and Dropout (p=0.1). |
| Submodular $\lambda$ | 1 | Balancing factors for concept selection. |

- **SOTA Adaptive Fusion:** For a fair comparison with *M4Survive*, which uses Mamba-based adapters, we implemented a similar adapter-based strategy. We inserted lightweight Mamba blocks to process the features $f_r$ and $f_h$ before a fusion block, and fine-tuned only the adapter weights, keeping the vision backbones frozen.

## A.4 ADAPTATION OF MOTCAT/PIBD TO RADIOLOGY–PATHOLOGY

Since MOTCAT and PIBD were originally proposed for genomics-pathology. We describe our adaptation details to radiology-pathology.

**MOTCAT.** The MOTCAT architecture features a unidirectional co-attention mechanism, where one modality serves as a "guidance" stream to refine the representation of the other. Since MOTCAT was originally designed for genomics guiding pathology, adapting it to our radiology-pathology setting required selecting a guidance direction. We evaluated both possible configurations: (1) Pathology-guides-Radiology, and (2) Radiology-guides-Pathology.

To determine the optimal configuration for our baseline, we conducted an empirical comparison on both the TCGA-GBMLGG and Center1-GC datasets for the cancer grading task. The results are summarized in Table 6. Our evaluation shows that using radiology as guidance yields slightly better performance while being more computationally efficient. Hence, we selected the Radiology-guides-Pathology configuration as the MOTCAT baseline for all experiments reported in the paper.

**PIBD.** We replace the (histology, genomics) pair with (pathology, radiology), project each to PIBD's shared width $D$, and pass the two embeddings to the original bottleneck/disentangling fusion. Objectives and downstream heads are unchanged.

Table 6: Comparison of MOTCAT performance with different guidance directions on the cancer grading task. We report mean ± standard deviation for AUC over 10 stratified splits.

| | TCGA-GBMLGG | | Center1-GC | |
|---|---|---|---|---|
| **Guidance Direction** | **AUC** ($\uparrow$) | **Time (h)** | **AUC** ($\uparrow$) | **Time (h)** |
| Pathology-guides-Radiology | $0.858 \pm 0.042$ | 4.32 | $0.638 \pm 0.046$ | 11.25 |
| Radiology-guides-Pathology | $0.865 \pm 0.025$ | 3.81 | $0.641 \pm 0.050$ | 9.13 |

## B DATASET DETAILS

We evaluate our framework on four datasets containing paired radiology and pathology images:

- **TCGA-LGG:** A cohort of 173 patients with Lower-Grade Glioma curated from The Cancer Genome Atlas (TCGA)[3]. For each patient, we obtained diagnostic whole-slide images (WSIs) and paired pre-operative, multi-parametric MRI scans (post-contrast T1-weighted and T2-FLAIR).

---

[3] https://www.cancer.gov/tcga/

Table 7: Summary of patient cohorts used for experiments.

| Dataset | Cancer Type | N | Survival Task | Grading Task | |
|---|---|---|---|---|---|
| | | | Censorship (%) | Task Description | # Classes |
| TCGA-LGG | Lower-Grade Glioma | 173 | 83.8% | WHO Tumor Grade | 3 |
| TCGA-GBM | Glioblastoma | 186 | 18.3% | | |
| Center1-GC | Gastric Cancer | 683 | 57.2% | TNM Stage (T-Stage) | 5 |
| Center2-CHS | Chondrosarcoma | 76 | — | WHO Tumor Grade | 5 |

- **TCGA-GBM:** Similarly, we curated a cohort of 186 patients with Glioblastoma Multiforme (GBM), the most aggressive primary brain tumor. The dataset consists of the same paired pre-operative MRI and WSI data types, matched with clinical survival outcomes from TCGA.

- **Center1-GC:** An in-house dataset of 683 gastric cancer patients. For each patient, we have a pre-operative CT scan and a post-resection WSI, acquired within one month of each other to ensure temporal consistency.

- **Center2-Chondrosarcoma:** An in-house dataset of 76 Chondrosarcoma patients with paired pre-operative MRI and WSI. Cohort characteristics are detailed in Table 7.

### B.1 ETHICAL CONSIDERATIONS AND DATA USAGE

All data from TCGA are publicly available and de-identified. The in-house datasets (Center1-GC, Center2-CHS) were collected under protocols approved by the local Institutional Review Board (IRB), with all patient data fully anonymized and de-identified prior to its use in this research. Informed consent was obtained from all participants included in the in-house studies. Our study strictly adheres to data privacy and protection regulations.

### B.2 PAIRING WINDOW AND EXCLUSION CRITERIA

We require fully paired radiology–pathology examples for all experiments; pairing is performed at the patient level (no spatial registration), and only cases satisfying the following rules are retained.

**In-house cohorts (Center1–GC, Center2–CHS).** We pair the pre–operative CT/MRI study with the diagnostic resection WSI from the same surgical episode. When multiple candidates exist, we select the imaging study and slide whose acquisition dates yield the minimum absolute time gap, and we require a pairing window of $|\Delta t| \leq 30$ days. If multiple WSIs are available, we use the slide annotated as "diagnostic" (primary tumor block).

**TCGA cohorts (LGG/GBM).** We pair MRIs and WSIs by TCGA case ID. When multiple scans/slides are available, we choose the pre–operative MRI closest in time to the diagnostic histology slide for that case. Pairs with missing or ambiguous identifiers/metadata are excluded.

## C CONCEPT SELECTION AND DETAILED CONCEPT LISTS

Our framework's interpretability is founded on a set of high-quality medical concepts. As described in the main paper (Section 3.1), we use a submodular optimization approach to select a set of concepts that are both prognostically relevant (high mutual information with survival outcome) and semantically diverse (low cosine similarity between embeddings). This approach aligns with recent trends in building more transparent models by grounding them in human-understandable concepts (Yamaguchi et al., 2025).

While Concept Bottleneck Models (CBMs) (Koh et al., 2020) pioneered this direction, they can suffer from performance degradation and instability, limiting their use in high-stakes medical applications (Hu et al., 2025). Our CTF framework avoids these pitfalls not by forcing information through a rigid bottleneck, but by using concepts as a dynamic, tunable semantic bridge, as demonstrated in our ablation studies.

## C.1 LLM Prompt Templates and Generation Pipeline

We use Gemini-2.5-pro to generate concept pools. This appendix documents the exact prompts, generation settings, and post-processing used to build the candidate concept pools in Sec. 3.1 from large language models (LLMs).

**Overview.** For each disease and modality, we query the LLM once per progression stage in S = early, intermediate, advanced, metastatic. Each query asks for N = 250 short, atomic features as a numbered list in the format "1.; 2.; 3.; ...". We then parse the enumeration, clean tokens, and save a JSON file keyed by stage: `"early": [...], "intermediate": [...], "advanced": [...], "metastatic": [...]` . Files are saved as `cancer_{modal}.json` with `modal` in {rad, path}.

**Prompt templates (exact text).** We use the following two templates, differing only by modality. Bracketed fields are filled programmatically.

> **Template R (Radiology, MRI)**
>
> What are the radiological features in Magnetic Resonance images of [CANCER] at the [STAGE] stage of progression (differentiate early, intermediate, advanced, metastatic stages), please describe using keywords or short sentences. Give [N] features and answer the question with the following format: 1.; 2.; 3.; ... .

> **Template H (Histopathology, WSIs)**
>
> What are the morphological features in Whole Slide Images (WSIs) of [CANCER] at the [STAGE] stage of progression (differentiate early, intermediate, advanced, metastatic stages), please describe using keywords or short sentences. Give [N] features and answer the question with the following format: 1.; 2.; 3.; ... .

**Generation settings.** Unless noted otherwise, we use default sampling parameters of the LLM-Backend (no explicit temperature or top-$p$ overrides). We issue one request per stage and modality-disease pair, with `N=250`. Radiology requests specify MRI/CT. Pathology requests specify H&E WSIs.

**Parsing and cleaning.** The LLM returns a numbered list. We extract items using an enumeration-aware regex:

```
^\s*\d+\.\s*(.*?)(?=\n\s*\d+\.|\Z)
```

applied with multiline and dotall flags. We then:

- trim whitespace and punctuation; remove leading bullets/asterisks; drop empty entries;
- cap at $N$ concepts per stage; preserve the original order;
- save to `cancer_{modal}.json` as a dictionary keyed by stage.

We subsequently merge the four stage lists, deduplicate (exact and fuzzy string match), and pass the merged pool to the MI + submodular selector (Sec. 3.1).

**Reproducibility and anonymity.** API keys are never embedded in the PDF or repository. We use environment variables (e.g., `GEMINI_API_KEY`, `OPENAI_API_KEY`).

**Post-processing.** We merge the four stage lists per (disease, modality), deduplicate by (i) exact match after lowercasing and punctuation stripping and (ii) fuzzy string match, then supply the cleaned pool to the mutual-information ranking and submodular diversity selection described in Sec. 3.1.

## C.2 GASTRIC CANCER (CENTER1-GC) CONCEPTS

Tables 8 and 9 list the top 30 (out of 256) selected concepts for the Center1-GC dataset for radiology and pathology, respectively.

Table 8: 30 selected Radiology concepts for the Center1-GC (Gastric Cancer) dataset.

| | | |
|---|---|---|
| 1. Poorly defined/irregular tumor margins | 11. Gastric outlet obstruction | 21. Venous encasement/invasion |
| 2. Marked heterogeneous enhancement | 12. Linitis plastica appearance | 22. Infiltration of adjacent organs |
| 3. Asymmetric or eccentric wall thickening | 13. Effacement of perigastric fat planes | 23. Peritoneal carcinomatosis |
| 4. Presence of tumor ulceration | 14. Definite serosal involvement | 24. Omental caking |
| 5. Tumor necrosis or necrotic core | 15. Tumor spiculation | 25. Distant metastasis to ovaries |
| 6. Large, matted regional lymph nodes | 16. Air within the tumor | 26. Enlarged Virchow's node |
| 7. Lymphatic spread to regional nodes | 17. Solid tumor component | 27. Visible feeding vessels (neovascularity) |
| 8. Presence of ascites | 18. Distortion of mucosal folds | 28. Arterial encasement |
| 9. Liver metastases with rim enhancement | 19. Moderate arterial phase enhancement | 29. Adrenal metastases |
| 10. Invasion into the muscularis propria | 20. Mass effect on adjacent structures | 30. Presence of tumor calcifications |

Table 9: 30 selected Pathology concepts for the Center1-GC (Gastric Cancer) dataset.

| | | |
|---|---|---|
| 1. Loss of glandular architecture | 11. High mitotic activity | 21. Tumor-infiltrating lymphocytes present |
| 2. Solid growth pattern | 12. Areas of tumor necrosis | 22. Tumor-associated macrophages present |
| 3. Presence of signet ring cells | 13. Increased nuclear-cytoplasmic ratio | 23. Epithelial-mesenchymal transition |
| 4. High-grade cellular atypia | 14. Prominent or irregular nucleoli | 24. High level of Microsatellite Instability |
| 5. Lymphovascular invasion (LVI) | 15. Discohesive cells presence | 25. Increased HER2 expression |
| 6. Perineural invasion | 16. Spindle cell morphology | 26. Loss of E-cadherin expression |
| 7. High tumor budding | 17. Desmoplastic reaction | 27. Increased Ki-67 proliferation index |
| 8. High tumor-stroma ratio | 18. Extracellular mucin pools | 28. Abnormal vessel morphology |
| 9. Poorly differentiated features | 19. Tumor cell apoptosis | 29. Increased Cyclin D1 expression |
| 10. Disruption of basement membrane | 20. Cribriform growth pattern | 30. Multinucleation / giant cells present |

## C.3 BRAIN TUMOR (TCGA-LGG/GBM) CONCEPTS

To demonstrate the adaptability of our concept selection strategy, Tables 10 and 11 list the top 30 selected concepts for the TCGA glioma cohorts. These concepts are distinct from those for gastric cancer and reflect the specific pathology of brain tumors.

Table 10: 30 selected Radiology concepts for the TCGA-LGG/GBM datasets, curated from the provided list.

| | | |
|---|---|---|
| 1. Central necrosis | 11. Subependymal spread | 21. Obstructive hydrocephalus |
| 2. Cystic components within the tumor | 12. New satellite lesions | 22. Uncal herniation |
| 3. Patchy enhancement | 13. Multifocal disease | 23. Compression of the brainstem |
| 4. Ill-defined or infiltrative margins | 14. Dissemination through CSF | 24. Superficial cortical involvement |
| 5. Increased tumor heterogeneity | 15. Breakdown of the blood-brain barrier | 25. Perfusion abnormalities |
| 6. Mass effect on adjacent structures | 16. New areas of restricted diffusion (DWI) | 26. Positive amino acid PET |
| 7. Compression of ventricles | 17. Lobulated appearance | 27. Small area of new enhancement |
| 8. Midline shift | 18. Elevated choline/creatine ratio (MRS) | 28. Increased perilesional edema |
| 9. Internal septations within cystic areas | 19. Decreased NAA (N-acetyl aspartate) | 29. Subtle new vascularity |
| 10. Deep gray matter structures | 20. Loss of gray-white matter differentiation | 30. Infiltration of tentorium |

Table 11: Top 30 selected Pathology concepts for the TCGA-LGG/GBM datasets. These concepts were selected from a larger pool to represent the core histopathological, molecular, and microenvironmental features of high-grade gliomas like Glioblastoma.

| | | |
|---|---|---|
| 1. Pseudopalisading necrosis | 11. Perineuronal satellitosis | 21. ATRX loss |
| 2. Microvascular proliferation | 12. EGFR amplification | 22. Lack of 1p/19q co-deletion |
| 3. High-grade cellular atypia | 13. TERT promoter mutation | 23. Hypoxia |
| 4. High mitotic activity | 14. MGMT promoter methylation status | 24. Tumor-associated macrophages |
| 5. Increased cellularity | 15. IDH1 mutation status | 25. M2-polarized macrophages |
| 6. Nuclear pleomorphism | 16. p53 mutations | 26. T-cell exhaustion |
| 7. Presence of multinucleated giant cells | 17. PTEN loss | 27. PD-L1 expression |
| 8. Glomeruloid bodies | 18. Chromosomal gains | 28. Aberrant GFAP expression |
| 9. Prominent nucleoli | 19. Chromosomal losses | 29. More expression of stem cell markers |
| 10. Diffuse infiltration of brain parenchyma | 20. Nuclear hyperchromasia | 30. Genomic instability |

## D    ADDITIONAL QUALITATIVE AND INTERPRETABILITY ANALYSIS

### D.1    PATHOLOGY ATTENTION HEATMAPS

Figure 7 presents a compelling comparison of attention maps from our model's pathology stream for representative patients from the Center1-GC cohort. The heatmaps visualize attention weights from the feature aggregation module, where red indicates regions receiving the highest attention. In the high-risk patient (b), the model correctly localizes its attention on dense, disorganized clusters of tumor cells characteristic of poorly differentiated carcinoma. Conversely, in the low-risk patient (a), the attention is sparse, indicating the absence of these aggressive features.

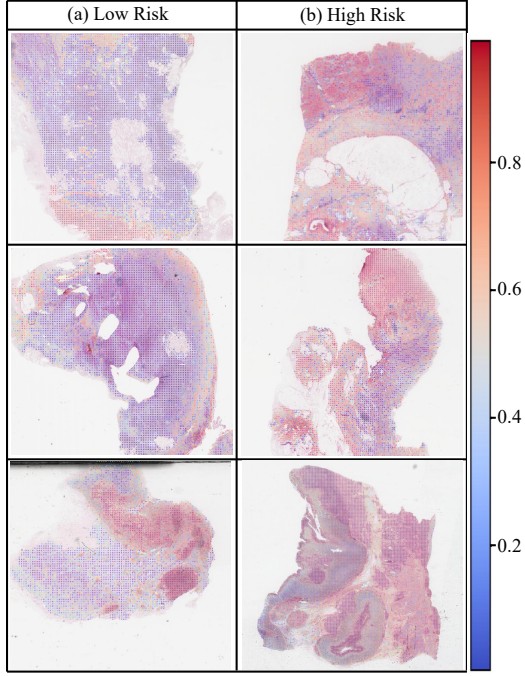

Figure 7: Pathology Attention Heatmaps for High- and Low-Risk Patients. The figure contrasts the model's spatial attention for (a) low-risk patients and (b) high-risk patients from the Center1-GC cohort.

### D.2    TIME-DEPENDENT AUC ANALYSIS

The Concordance Index (C-index) provides a single, global measure of a model's rank-based discriminatory ability. However, in survival analysis, a model's predictive accuracy can vary over time. The time-dependent Area Under the Curve (td-AUC) offers a more granular evaluation by assessing the model's ability to distinguish patients who will experience an event before a specific time point t from those who will not (Schmid et al., 2015).

As shown in Figure 8, CTF demonstrates not just a superior but a strikingly dominant performance over the PIBD baseline on the Center1-GC dataset. CTF maintains a high td-AUC (consistently > 0.75) across the entire follow-up period, achieving an excellent mean AUC of 0.808. In stark contrast, the PIBD baseline performs at a level comparable to random chance, with a mean AUC of 0.483. This significant and persistent performance gap, highlighted by the shaded green area, indicates that while the latent-space fusion model fails to maintain discriminative power over time, CTF's dynamic, concept-based fusion provides robust and reliable prognostic predictions for both near-term and longer-term outcomes.

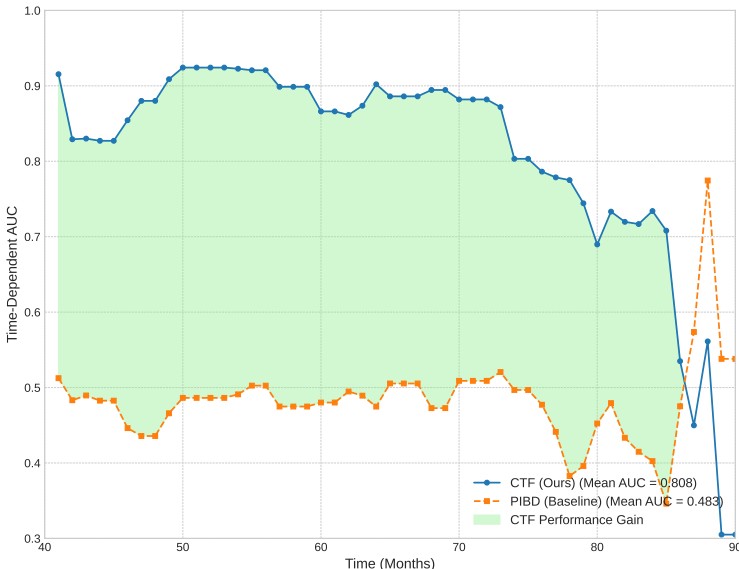

Figure 8: Time-dependent AUC curves on the Center1-GC dataset comparing CTF to the PIBD baseline. CTF (blue) consistently and significantly outperforms the baseline across all time horizons, maintaining robust prognostic accuracy. The shaded green area highlights the large performance gain.

Table 12: Concept intervention analysis on predicted risk scores. For two representative patients from the Center1-GC dataset, we intervene on the highest-scoring concepts by clamping their score to 0.0 and observing the change in the final predicted risk. The percentage change ($\Delta$ Risk) highlights the causal influence of each concept. $C$ and $R_n$ represent original concept scores and new risk scores.

| Intervened Concept | Modality | $C$ | $R_n$ | $\Delta$ Risk (%) |
|---|---|---|---|---|
| **Patient A: High-Risk** (*Actual Outcome: Event at 25 months*) **Initial Predicted Risk: 3.91** | | | | |
| Bad blood vessel morphology | Radiology | 3.35 | 2.98 | **-23.8%** |
| Lymphovascular invasion | Pathology | 2.80 | 3.02 | **-22.7%** |
| Disseminated tumor cells | Pathology | 2.00 | 3.22 | **-17.6%** |
| **Patient B: Low-Risk** (*Actual Outcome: Censored at 54 months*) **Initial Predicted Risk: -2.57** | | | | |
| Well-differentiated | Pathology | 1.65 | -1.94 | **+24.5%** |
| Circumscribed tumor margins | Radiology | 1.20 | -2.18 | **+15.2%** |
| Low tumor-stroma ratio | Pathology | 0.85 | -2.37 | **+7.9%** |

## D.3 CONCEPT INTERVENTION

**Concept Intervention.** To test if the learned concepts have a causal impact on predictions, we performed concept intervention experiments by neutralizing high-impact concept scores for representative patients. As shown in Table 12, intervening on concepts like "Bad blood vessel morphology" for a high-risk patient or "Well-differentiated" for a low-risk patient resulted in significant and clinically plausible shifts in the final risk score (a 23.8% decrease and 24.5% increase, respectively). This provides strong evidence that CTF's predictions are causally linked to interpretable concepts.

## D.4 STATISTICAL SIGNIFICANCE TESTS

For each dataset and metric, we compare CTF against the strongest competing baseline using a paired t-tests across the 10 stratified splits. We report raw p-values in Table 13 below.

Table 13: Paired one-sided $t$-test (over 10 stratified splits) comparing **CTF** with the strongest competing multimodal baseline on each dataset and task. Reported are $p$-values for the null hypothesis that there is no difference in mean performance between CTF and the baseline.

| Task | Dataset | Metric | CTF | Best baseline | $p$-value |
|------|---------|--------|-----|---------------|-----------|
| Survival | TCGA-LGG | C-index | $0.713 \pm 0.103$ | M4Survive ($0.709 \pm 0.112$) | 0.13 |
| | TCGA-GBM | C-index | $0.579 \pm 0.063$ | MOTCAT ($0.563 \pm 0.108$) | 0.09 |
| | Center1-GC | C-index | $0.665 \pm 0.061$ | M4Survive ($0.642 \pm 0.065$) | 0.05 |
| Grading (AUC) | TCGA-GBMLGG | AUC | $0.903 \pm 0.028$ | Cross-Attention ($0.868 \pm 0.030$) | 0.02 |
| | Center2-CHS | AUC | $0.854 \pm 0.081$ | M4Survive ($0.830 \pm 0.075$) | 0.06 |
| | Center1-GC | AUC | $0.660 \pm 0.049$ | M4Survive ($0.649 \pm 0.052$) | 0.05 |
| Grading (ACC) | TCGA-GBMLGG | ACC | $0.718 \pm 0.063$ | M4Survive ($0.691 \pm 0.061$) | 0.05 |
| | Center2-CHS | ACC | $0.698 \pm 0.164$ | M4Survive ($0.626 \pm 0.115$) | 0.02 |
| | Center1-GC | ACC | $0.401 \pm 0.057$ | Cross-Attention ($0.394 \pm 0.049$) | 0.06 |

# E   BROADER IMPACT AND LIMITATIONS

## E.1   POTENTIAL FOR POSITIVE IMPACT

The successful development and deployment of the CTF framework could have a significant positive impact on clinical oncology and computational medicine.

1. **Improved Prognostic Accuracy:** By creating a deeper synergy between radiology and pathology, CTF can provide more accurate and reliable predictions of patient outcomes. This could help clinicians better stratify patients for treatment, identifying high-risk individuals who may benefit from more aggressive therapies and low-risk individuals for whom de-escalation could be considered.

2. **Enhanced Clinical Decision Support:** The interpretable nature of CTF is a key advantage. By presenting predictions alongside the contributing medical concepts (e.g., "High score for 'Lymphovascular invasion'"), the model can serve as a "second-read" tool that not only provides a risk score but also highlights the key evidence, facilitating a more informed dialogue between the AI and the clinician.

3. **Accelerated Scientific Discovery:** The cross-modal influence analysis (Figure 6) can uncover novel or subtle correlations between macroscopic imaging features and microscopic cellular patterns. This could generate new hypotheses for translational research into the biological drivers of cancer aggression.

## E.2   LIMITATIONS AND FUTURE WORK

Despite its promising results, our work has several limitations that open avenues for future research. A primary limitation is the framework's dependence on the quality and comprehensiveness of the initial concept vocabulary. The performance of CTF is fundamentally tied to the concepts provided, and while we used LLMs to generate a broad list, this process may miss crucial niche concepts or introduce biases. Future work should therefore explore more robust, data-driven methods for concept discovery or involve domain experts in a formal human-in-the-loop process to refine and validate the concept library.

Furthermore, our framework leverages powerful, pre-trained Vision-Language Models, inheriting both their extensive knowledge and their potential biases. The model's performance is ultimately capped by the quality of these foundational backbones. While our ablation study confirmed that expert models perform best, a valuable future direction would be to investigate methods for jointly fine-tuning the prompt modules and a small fraction of the backbone model's weights to achieve even better task-specific adaptation. This must be balanced against computational costs, as the forward pass for CTF, though parameter-efficient in its training, remains resource-intensive due to the multiple large models and the WSI feature extraction bottleneck. Further optimization would be required for real-time clinical deployment.

Perhaps the most significant barrier to immediate clinical translation is the model's requirement for fully-paired data—one radiology and one pathology image per patient—during training. This

is a considerable constraint, as real-world clinical datasets are often incomplete. Extending CTF to gracefully handle missing modalities is therefore a critical next step. Future iterations could investigate learning to impute concept scores from the available modality or using dynamic attention mechanisms to operate effectively even with an incomplete data stream.

## F  STATEMENT ON THE USE OF LARGE LANGUAGE MODELS (LLMS)

We disclose the use of Large Language Models (LLMs) in this work. LLMs played a role in the following ways:

1. **Concept Generation:** As detailed in Section 3 and Appendix C.1, we utilized an LLM to generate an initial broad pool of candidate radiological and pathological concepts. This served as a starting point for our prognostic and diversity-based concept selection algorithm.

2. **Language Polishing:** LLMs were used as a general-purpose writing assistant to improve the clarity, grammar, and style of the manuscript.

