# OpenReview forum: "Bridging Radiology and Pathology Foundation Models via Concept-Based Multimodal Co-Adaptation"
_ICLR.cc/2026/Conference — ICLR 2026 Poster_

### Official Review · Reviewer_N7n7 · 2025-10-27

**Soundness:** 2
**Presentation:** 2
**Contribution:** 2
**Rating:** 4
**Confidence:** 5

**Summary:**

The authors proposed a feature alignment scheme for bridging the radiology and histopathology image foundation models. A list of concepts is first selected and then used as the linkage feature between radiology and histopathology image features, with separate adaptors trained. The performance evaluation is conducted on two downstream tasks, i.e., survival analysis and tumor grading. Three datasets (two public and one private) are adopted in the experiments, and only marginally improved results of the proposed method are reported in comparison to previous multi-modal feature fusion methods, e.g., 1.6-2.6% increase in C-index for survival analysis. The manuscript is not easy to follow overall. Many details seem to be missing from the introduction, along with several flaws detailed below.

**Strengths:**

+ The paper proposed a multi-modal alignment scheme that required light tuning efforts, which could be easily adopted in the downstream tasks.
+ The downstream in the evaluation is clinically relevant.

**Weaknesses:**

- There are many details of the proposed methods missing. 1. It is not clear how the 3D volumes of radiology images and WSI
 Images are encoded. It could largely influence the semantics of the image embedding, especially for the alignment task. 2. Why is the context prompt only computed for the histopathology images? 3. Details of the MoE layer? 4. Is P_tuned computed for all the concepts in both sets? 5. How are the images associated with the concepts? 6. Are there any other inputs other than f_r and f_h?
- The overall lack of clarity in the introduction makes the paper a bit hard to follow.
- Radiologists and histopathologists speak quite different languages, which is also reflected by the selected concepts shown in the appendix. How often can these concepts be aligned, and it will be helpful to show what percentage of these concepts are matched.
- The proposed method utilized the global features of the images, while the concepts are composed in more detail. How will this mismatch of the semantic feature levels affect the performance? How different could the features be after the adoption in comparison to the original image feature? It is also related to the following point.
- The overall performance gain after adopting the proposed method is marginal, e.g.,1.6-2.6% increase in C-index for survival analysis in comparison to the recent baseline. Is the performance gain sourced from the additionally concept injected? What will be the performance using solely the two image features and textual features (e.g., simple concatenation)
-

**Questions:**

see weakness

---

> ### Author Response · Authors · 2025-11-22
> **Response to Reviewer N7n7 (Part 1 of 3)**
>
> We thank the reviewer for the detailed feedback and for carefully going through our manuscript.
>
> >**[W1]** There are many details of the proposed methods missing. 1. It is not clear how the 3D volumes of radiology images and WSI Images are encoded. It could largely influence the semantics of the image embedding, especially for the alignment task. 2. Why is the context prompt only computed for the histopathology images? 3. Details of the MoE layer? 4. Is P_tuned computed for all the concepts in both sets? 5. How are the images associated with the concepts? 6. Are there any other inputs other than f_r and f_h?
>
> Per your suggestion, we have clarified all raised implementation details as follows.
>
>
> **1. Encoding of WSIs and radiology volumes.**
> We have added a full description of our encoding pipeline to **Appendix A.2**.
> We summarize it here:
>
> For WSIs (pathology), each whole‑slide image is processed at 20× magnification, tiled into 256×256 patches, foreground‑filtered (Otsu + tissue mask), and embedded with the CONCH vision encoder. We then use an attention‑based MIL aggregator to obtain a single global pathology feature vector $\\boldsymbol{f}_h$ per patient.
>
> For radiology volumes, on our private datasets, we used 2D axial slices from expert-segmented tumor regions. For the public TCGA cohorts, we used the entire axial slice with the largest tumor cross-section. We adopted a 2D encoder (BiomedCLIP) as it empirically outperformed 3D models on our datasets, which include CT volumes with highly anisotropic voxel spacing. To further validate this choice, we provided an additional analysis on the Center1-GC dataset, summarized below (**Table R4.1**).
>
> **Table R4.1: Radiology-only survival prediction on Center1-GC (C-index↑) using different radiology encoders**
>
> | Dataset | 2D BiomedCLIP | 3D Merlin [1] | 3D CT-CLIP [2] |
> |---|---:|---:|---:|
> | Center1-GC | **0.614±0.052** | 0.568±0.078 | 0.583±0.056 |
>
> **2. Symmetry of Context Prompts.**
> This is a misunderstanding caused by our figure/text, and we apologize for the confusion.
> In our method, the context prompt is defined symmetrically for both modalities.
> For a radiology concept $c_r$, the context prompt $\\boldsymbol{P}_C\\left(c_r,\\boldsymbol{f}_h\\right)$ is generated from the pathology feature $\\boldsymbol{f}_h$.
> For a pathology concept $c_p$, the context prompt $\\boldsymbol{P}_C\\left(c_p,\\boldsymbol{f}_r\\right)$ is generated from the radiology feature $\\boldsymbol{f}_r$.
> We revised **Figure 2** to show both directions and clarified wording explicitly in **Section 3.2**.
>
> **3. MoE layer details.**
> We added a more explicit description of the MoE‑style context prompt generator in **Section 3.2 and Appendix A.1**.
> Concretely, for radiology concepts (pathology is symmetric):
> - We maintain a pool of $M$ basis prompts $\\{\\boldsymbol{P}^{\\text{basis}} _ {C,i}\\}_{i=1}^M$ (learnable tensors).
> - A small gating network $g_r: \\mathbb{R}^{D_{vh}} \\to \\mathbb{R}^M$ takes the complementary modality feature $\\boldsymbol{f}_h$ and outputs logits $\\boldsymbol{\\beta}=g_r\\left(\\boldsymbol{f}_h\\right)$.
> - We apply a softmax to obtain mixture weights $\\alpha=\\text{softmax}(\\boldsymbol{\\beta})$.
> - The context prompt is then $\\boldsymbol{P} _ {C}(\\boldsymbol{f} _ h) = \\sum_{i=1}^{M} \\alpha _ i \\cdot \\boldsymbol{P}^{\\text{basis}} _ {C,i}$.
>
> **4. $\\boldsymbol{P}_{tuned}$ for all concepts?**
> Yes. For each selected concept in each modality, $\\boldsymbol{P}_{\\text{tuned}}(c) = \\text{Concat}(\\boldsymbol{P}_G, \\boldsymbol{P}_C(\\cdot), \\boldsymbol{P}_S(\\cdot))$ is prepended to the tokenized concept string before the frozen text encoder.
>
> ****Reference****
>
> [1] Blankemeier, et al. Merlin: A vision language foundation model for 3d computed tomography. Research Square (2024).
>
> [2] Hamamci, et al. Developing generalist foundation models from a multimodal dataset for 3d computed tomography. arXiv (2024).

---

> ### Author Response · Authors · 2025-11-22
> **Response to Reviewer N7n7 (Part 2 of 3)**
>
> **(Cont’) W1.5 How images associate to concepts.**
> There are two stages in which images are associated with concepts:
>
> The first stage is \textbf{concept selection (MI step)}. Initially, for each candidate concept $c$ and image $x_i$ in a given modality, we compute an initial alignment score $a(\\boldsymbol{x}_i, c) = \\boldsymbol{t}_c^\\top\\boldsymbol{f}_i/(\\|\\boldsymbol{t}_c\\|_2 \\|{\\boldsymbol{f}}_i\\|_2)$, where $\\boldsymbol{f}_i$ is the frozen vision feature of sample $x_i$ and $\\boldsymbol{t}_c$ is the frozen text embedding of concept $c$.
> Concepts are ranked and selected by the mutual information between $a(\\boldsymbol{x}, c)$ and label.
>
> The second stage is in the **final model (tuned concepts)**. During training and inference with CTF, each tuned concept embedding $\\tilde{t}_c$ is aligned with its modality-specific vision feature via cosine similarity:
> $$
>  \\boldsymbol{s} _ {r,j} = \\frac{\\boldsymbol{f} _ r^\\top \\tilde{\\boldsymbol{t}} _ {c^{(j)} _ r}}{\\lVert \\boldsymbol{f} _ r \\rVert _ 2 \\, \\lVert \\tilde{\\boldsymbol{t}} _ {c^{(j)} _ r} \\rVert _ 2},
>  \\boldsymbol{s} _ {h,j} = \\frac{\\boldsymbol{f} _ h^\\top \\tilde{\\boldsymbol{t}} _ {c^{(j)}_h}}{\\lVert \\boldsymbol{f} _ h \\rVert _ 2 \\, \\lVert \\tilde{\\boldsymbol{t}} _ {c^{(j)} _ h} \\rVert _ 2},
> $$
> which yields the concept-score vectors $\\boldsymbol{s}_r$ and $\\boldsymbol{s}_h$ for radiology and pathology.
> These scores are concatenated and fed into the prediction head.
> In this way, each image is associated with each concept via cosine similarity between the image feature and the (tuned) concept embedding.
>
> **W1.6. Are there any other inputs other than $f_h$ and $f_r$?**
> No. At inference time, the network’s inputs are only the radiology image (from which we derive $f_r$), and the pathology WSI (from which we derive $f_h$).
> Clinical labels are used only in the loss and in the offline mutual‑information calculation for concept selection.
>
> >**[W2]** The overall lack of clarity in the introduction makes the paper a bit hard to follow.
>
> Thank you for pointing this out. We have \textbf{revised the Introduction} to make the motivation and pipeline easier to follow. We added a clear contrast of "static latent fusion" vs. "dynamic concept co‑adaptation"  and a 3‑sentence roadmap outlining:  (i) MI + diversity concept selection (Sec. 3.1), (ii) GCSP‑based cross‑domain concept tuning (Sec. 3.2), and (iii) concept‑score fusion and prediction (Sec. 3.3)
>
> >**[W3]** Radiologists and histopathologists speak quite different languages, which is also reflected by the selected concepts shown in the appendix. How often can these concepts be aligned, and it will be helpful to show what percentage of these concepts are matched.
>
> This is an excellent point. Our framework does not enforce a hard 1-to-1 alignment between the disparate concept vocabularies. Instead, the GCSP mechanism facilitates a "soft," contextual alignment by allowing concepts in one domain to be dynamically influenced by features from the other.
>
> To quantify emergent cross‑modal correspondence, we added a new analysis here. For each radiology concept, we compute its best‑matching pathology concept via cosine similarity between their embeddings under the same encoder. As shown in **Table R4.2**, the concepts have **moderate** semantic alignment, with 46% of radiology concepts having a pathology counterpart with a cosine similarity of at least 0.6.
>
> However, as the reviewer implies, a simple post-hoc similarity score is just the beginning. The reason this is such a challenging problem is that the relationship is **rarely one-to-one** and a single radiological finding may result from **a complex interplay** of multiple microscopic processes. Therefore, building a more explicit and deeper alignment between these domains is a crucial direction for future work. We envision two main avenues: (1) cross-modal knowledge graph: constructing a knowledge graph to explicitly model biological links between concepts, creating a transparent and verifiable reasoning framework for clinicians; and (2) unsupervised concept alignment: developing methods to automatically discover aligned concepts by learning correlated patterns directly from imaging data, enabling the discovery of novel biomarkers.
>
> **Table R4.2: Cross‑modality concept correspondence on Center1‑GC. For each radiology concept, we find its best‑matching pathology concept and report the proportion of pairs whose cosine similarity exceeds a given cosine similarity threshold**
>
> | | $\\Pr[\\cos\\ge 0.5]$ | $\\Pr[\\cos\\ \\ge 0.6]$ | $\\Pr[\\cos\\ \\ge 0.7]$ |
> |---|---:|---:|---:|
> | Proportion | 0.61 | 0.46 | 0.27 |

---

> ### Author Response · Authors · 2025-11-24
> **Response to Reviewer N7n7 (Part 3 of 3)**
>
> >**[W4]** The proposed method utilized the global features of the images, while the concepts are composed in more detail. How will this mismatch of the semantic feature levels affect the performance? How different could the features be after the adoption in comparison to the original image feature? It is also related to the following point.
>
> This is a crucial point regarding semantic consistency. The potential mismatch between a global image feature and detailed concepts is a valid concern. Our design addresses this in two key ways.
>
> First, the “global” pathology feature $\\boldsymbol{f}_h$ is not a simple average but an **attention-weighted summary** of thousands of microscopic patch embeddings. The model learns to focus on the most diagnostically salient regions, so the resulting global vector is inherently imbued with the concept-level information from those critical areas. This attention mechanism significantly mitigates the risk of a severe semantic mismatch.
>
> Second, to further bridge any remaining gap, our GCSP mechanism dynamically adapts the concept embeddings themselves. To quantify how this tuning process alters semantics, we conducted a ``semantic drift'' analysis, which is now included in **Section 4.5 (Figure 6)**. We found that the tuned embeddings remain moderately similar to the original ones (median cosine similarity of 0.633), indicating that our model performs a task-specific **sharpening** of concepts rather than a wholesale redefinition that would destroy interpretability.
>
> Finally, this clinically coherent refinement is further validated by our concept intervention study (**Appendix D.3, Table 11**), which demonstrates that the tuned concept scores have a clinically plausible influence on the model's final prediction.
>
>
> >**[W5]** The overall performance gain after adopting the proposed method is marginal, e.g.,1.6-2.6% increase in C-index for survival analysis in comparison to the recent baseline. Is the performance gain sourced from the additionally concept injected? What will be the performance using solely the two image features and textual features (e.g., simple concatenation)
>
> We thank you for raising this important question about the source and significance of our performance gains. While the absolute improvements may seem modest, in the context of survival analysis, C-index gains of 2%–4% are often considered clinically meaningful [3].
>
> We are encouraged that our method achieves these consistent gains across multiple diverse cohorts. To rigorously validate these improvements, we have performed paired t-tests over 10 splits, and the results (now in **Appendix D.4, Table 12**) show that several of these gains are statistically significant (e.g., p$\\le$0.05 on Center1-GC AUC and TCGA-GBMLGG AUC/ACC).
>
> To address your question about whether the gain is simply from injecting concepts, our ablation studies in **Table 3 (also shown in Table R4.3)** provide a clear answer. The gain is not from adding concepts as static features, but from our dynamic co-adaptation mechanism. For instance, the “Static Concepts (CBM)” variant, which uses a fixed concept bottleneck, performs very poorly (C-index drop of -0.079). Besides, removing only the cross-domain **Context Prompt ($\\boldsymbol{P}_C$)**—the core of our co-adaptation strategy—causes the single largest performance drop. This directly demonstrates that the key advantage comes from the dynamic, cross-modal dialogue that GCSP enables, rather than the mere presence of concepts.
>
> Furthermore, we directly tested a simple fusion baseline as you suggested. Our “Concat-Fusion” model, which simply concatenates the image features $\\boldsymbol{f}_r$ and $\\boldsymbol{f}_h$ before feeding them to an MLP, performs notably worse than CTF (e.g., C-index of 0.626 vs. 0.665 on Center1-GC). Taken together, these results confirm that CTF's advantage is not derived from merely adding more features, but from its fundamental design: enabling an early, concept-level co-adaptation between modalities before they are fused for prediction.
>
> **Table R4.3: Ablation Recap for Weakness 5 (Center1–GC). Mean ± sd over 10 splits. $\\Delta$ is absolute change vs. CTF**
>
> | | Survival (C–index) ↑ | | Grading (AUC) ↑ | |
> |---|---:|---:|---:|---:|
> | Variant | mean±sd | Δ vs. CTF | mean±sd | Δ vs. CTF |
> | CTF (Full, GCSP) | **0.665 ± 0.061** | — | **0.660 ± 0.049** | — |
> | w/o Context Prompt ($P_{\\mathrm{C}}$) | 0.629 ± 0.058 | −0.036 | 0.635 ± 0.047 | −0.025 |
> | Concat–Fusion ($\\boldsymbol{f}_r, \\boldsymbol{f}_h\\to$ MLP) | 0.626 ± 0.048 | −0.039 | 0.629 ± 0.051 | −0.031 |
>
> ****Reference****
>
> [3] Schmid, et al. On the validity of time-dependent AUC estimators. Briefings in Bioinformatics (2015).

---

> > ### Comment · Reviewer_N7n7 · 2025-11-25
> >
> > I thank the authors for the detailed response. Most of the concerns are clarified in the rebuttal; therefore, I will raise my rating to lean towards acceptance.

---

> > > ### Author Response · Authors · 2025-11-25
> > >
> > > We sincerely thank Reviewer N7n7 for the continued engagement and thoughtful feedback. We are very pleased to hear that our rebuttal and the modifications to our manuscript have clarified your initial concerns.
> > >
> > > Your constructive comments have been invaluable in improving the quality and clarity of our paper, and we are very grateful for your willingness to raise your rating.

---

### Official Review · Reviewer_LEUR · 2025-10-28

**Soundness:** 2
**Presentation:** 3
**Contribution:** 3
**Rating:** 6
**Confidence:** 4

**Summary:**

This paper presents CTF (Concept Tuning and Fusing), a framework that bridges radiology and pathology foundation models using clinically meaningful concepts as an interpretable interface. Instead of fusing static features, CTF employs a Global-Context-Shared Prompt mechanism that dynamically adapts concept interpretations based on cross-domain information, allowing each modality to inform the other's concept understanding. Tested on survival prediction and cancer grading across multiple datasets, CTF achieves state-of-the-art performance while training only 0.15% additional parameters.

**Strengths:**

1.	Novel methodology: The Global-Context-Shared Prompt (GCSP) strategy dynamically conditions concept interpretations in one modality based on features from the complementary modality, enabling deeper synergy than traditional static feature fusion approaches.

2.	Parameter efficiency with strong performance: CTF achieves state-of-the-art results while requiring only 0.15% additional trainable parameters by keeping foundation models frozen and learning only lightweight prompt modules.

**Weaknesses:**

1.	The citations in this paper are **all incorrect**. Please correctly use the \citet and \citep LaTeX commands to distinguish proper references for different scenarios.

2.	Baselines such as PIBD, MOTCAT are originally developed for the fusion of pathological and genomic data. The comparison might be unfair for pathology and radiology data as inputs.

3.	Task “Center-1 GC” in Table 2 shows relatively low performance (all models) in AUC and especially ACC, suggesting this is not a proper task to choose for model evaluation.

**Questions:**

1.	Could the authors elaborate more on the registration of WSIs and radiology data, such as CT scans? How to make sure the data are paired since they are in different macro- and micro-scales?

2.	For the computation of the alignment score for concepts, why is it not normalized or scaled? Meanwhile, it’s then discretized into a variable, but how? Could the authors elaborate more, or specify where the reader can find related details in the Appendix? Moreover, since here the model requires the label to compute the mutual information, what’s the strategy during inference where there is no patient label?

3.	In Section 3.3, the authors mentioned that the concept score vectors “provide an interpretable representation of the patient’s condition”. Could the authors elaborate more on this conclusion?

4.	CTF trains in an end-to-end manner. Although most of the parameters of foundation models are frozen, this still introduces concerns about efficiency. Could the authors provide the least requirement to train CTF compared to uni-modal or late fusion baselines (which, as far as I know, can be trained in one GTX 3090 card)? Also, how many GPUs are used for the actual model training? (Appendix only mentions types of GPU)

5.	Selection strategy for pathology VLMs. It seems CONCH contributes largely to the performance of CTF from Table 3. But this ablation study only compares CONCH with the general-purpose CLIP.  What about other pathology VLMs such as PLIP [1] and MUSK [2]?

$\quad$ [1] Huang, Zhi, et al. "A visual–language foundation model for pathology image analysis using medical twitter." Nature medicine 29.9 (2023): 2307-2316.

$\quad$ [2] Xiang, Jinxi, et al. "A vision–language foundation model for precision oncology." Nature 638.8051 (2025): 769-778.

6.	Could the author please add these papers in the Introduction or Related Work sections since they are relevant:

$\quad$ [1] (Pathology Foundation Model) Chen, Richard J., et al. "Towards a general-purpose foundation model for computational pathology." Nature medicine 30.3 (2024): 850-862.

$\quad$ [2] (Pathology Foundation Model) Ma, Jiabo, et al. "A generalizable pathology foundation model using a unified knowledge distillation pretraining framework." Nature Biomedical Engineering (2025): 1-20.

$\quad$ [3] (Pathology Foundation Model) Xu, Hanwen, et al. "A whole-slide foundation model for digital pathology from real-world data." Nature 630.8015 (2024): 181-188.

$\quad$ [4] (Foundation Model Adaptation / Prompt Generation and Selection) Guo, Zhengrui, et al. "Focus: Knowledge-enhanced adaptive visual compression for few-shot whole slide image classification." Proceedings of the Computer Vision and Pattern Recognition Conference. 2025.

$\quad$ [5] (Foundation Model Adaptation / Parameter-efficient learning) Che, Haoxuan, et al. "Llm-driven medical report generation via communication-efficient heterogeneous federated learning." IEEE Transactions on Medical Imaging (2025).

**I will definitely consider raising my score if authors solve my concerns above.**

---

> ### Author Response · Authors · 2025-11-22
> **Response to Reviewer LEUR (Part 1 of 2)**
>
> We thank the reviewer for the detailed and constructive comments. Below we respond point-by-point.
>
> **W1.** **Incorrect citation formatting**
>
> Thank you very much for your suggestion on improving our reference format.
> We have rigorously replaced inline citations with the appropriate commands in our revised manuscript.
>
> **W2.** **Fairness of PIBD and MOTCAT as baselines**
>
> **Rationale.**
> Dedicated radiology–pathology fusion methods are scarce, and they often use simple fusion such as concatenation or cross-attention, which we have included.
> As the core contribution of PIBD and MOTCAT lies in the multimodal fusion within the embedding space, which is agnostic to the specific input modality,
> we believe that preserving their architectures and only substituting the input embeddings constitutes a fair and meaningful comparison.
>
> **Implementation.** All multimodal baselines (including PIBD/MOTCAT) and CTF use the same frozen vision encoders.
> For MOTCAT we kept its unidirectional design (primary: pathology; guidance: radiology) and injected radiology in place of genomics tokens.
> For PIBD we projected both streams to its bottleneck width and used the original disentangling/bottleneck fusion.
> We revised our Sec. 4.2 and Appendix A.3/A.4 accordingly.
>
> **W3.** **Low absolute performance on Center1‑GC grading (Table 2)**
>
> **Context and Rationale.** Center1‑GC is a 5‑way pre‑operative T‑stage task in a real‑world, class‑imbalanced cohort. Subtle classes (e.g., T3 vs T4) are intrinsically difficult from imaging alone, which depresses absolute ACC/AUC for all methods.
> However, Pre‑operative T‑stage estimation remains clinically relevant for surgical planning and neoadjuvant therapy.
>
> **Empirical Results.** Within this difficult scenario, our method still achieves significant improvements over all baselines in both AUC and ACC on this hardest task (Table 2), which we view as evidence of robustness. We also include this dataset because paired radiology–pathology cohorts are rare, and incorporating it allows for a more comprehensive and realistic evaluation of cross-modal fusion methods.
>
> **Q1. Registration/pairing of WSIs and radiology**
>
> **MRI registration.** For preprocessing, we followed the standard BraTS challenge pipeline, including file format conversion, skull stripping, and registration to a common anatomical template [1].
>
> **Data pairing.** Radiology and pathology images differ significantly in spatial scale and field of view, making direct spatial registration extremely challenging.
> Therefore, in our setting, we pair radiology and pathology images at the study level using patient ID, acquisition date, and anatomical site.
> This procedure ensures that the paired images share consistent clinical semantics, which is appropriate for our multimodal evaluation.
> Such study-level alignment provides a reliable basis for tuning our paired foundation models.
> We adjusted Sec. 4.1, and added the exact rules to Appendix B.2.
>
> **Q2. Alignment score, normalization, discretization, and labels vs inference**
>
> **Formula Correction.** For concept selection, we compute a cosine‑normalized alignment score
> $a(x_i, c) = \\boldsymbol{t}_c^\\top\\boldsymbol{f}_i/(\\|\\boldsymbol{t}_c\\|_2 \\|{\\boldsymbol{f}}_i\\|_2)$ (we apologize that we forgot to add normalization terms in our original formula for alignment scores) using the frozen encoders.
>
> **Discretization.** Because MI requires compatible variable types, we estimate the discrete–continuous MI via sklearn’s kNN‑based *mutual_info_classif* [2], which internally discretizes the continuous alignment scores (See [3] for more details) to be a discrete variable. We clarified this in Sec. 3.1 and cite the exact estimator.
>
> **Label Usage.** Labels are used only offline for concept selection. At inference, the model computes tuned concept embeddings, cosine concept scores, and the prediction head outputs a label‑free prediction.
>
> ****Reference****
>
> [1] https://github.com/neuronflow/BraTS-Toolkit.
>
> [2] Ross, B. C. (2014). Mutual information between discrete and continuous data sets. PloS one.
>
> [3] https://scikit-learn.org/stable/modules/generated/sklearn.feature_selection.mutual_info_classif.html.

---

> ### Author Response · Authors · 2025-11-22
> **Response to Reviewer LEUR (Part 2 of 2)**
>
> **Q3. Why concept score vectors are interpretable**
>
> The interpretability arises at several levels:
>
> Firstly, the interpretability lies in the semantic structure.
> Each dimension of the concept score vector corresponds to a clinically meaningful concept (e.g., “lymphovascular invasion”, “solid growth pattern”, “irregular tumor margins”).
> The value is a normalized similarity between the patient’s image feature and the tuned concept embedding.
> Hence, a large positive score is interpretable as “this concept is strongly present/invoked in this patient’s imaging”.
>
> Besides, we have patient‑level rationales.
> In Fig. 4, top concepts for low‑ vs high‑risk patients (e.g., low risk dominated by “well‑differentiated”, high risk dominated by “disseminated tumor cells”, “bad vessel morphology”), which align well with clinical expectations.
> In Table 11 (Appendix D), we perform concept interventions: clamping high‑impact concepts (e.g., “bad blood vessel morphology”) moves risk predictions in a clinically plausible direction, demonstrating the causal influence of concept scores.
>
> We also investigate cross‑modal influence.
> Fig. 7 summarizes how radiology concept groups modulate pathology concept scores via the context prompt $\\boldsymbol{P}_{C}$.
> The learned influence patterns (e.g., “invasion & metastasis” in radiology, amplifying “vascular invasion” and “cellular atypia” in pathology) correspond closely to how clinicians reason across modalities.
>
> **Q4. Efficiency and minimal hardware requirements**
>
> From our experience, the main memory driver is sequence length (set by the tokenizer) and the number of concepts per domain.
> All of our experiments were run on a single NVIDIA L40s GPU (48GB), and we recommend (sequence length, number of concepts per domain) to be (128, 256) respectively, as a default setting.
>
> However, we found out that the setting of (128, 256) could not be run on a single GTX 3090 card.
> The (128, 128) or (64, 256) or (128, 256/128, one domain has 256 concepts, the other one has 128 concepts) setting is a safe setting on a single GTX 3090, but the performance is slightly compromised (Table R3.1).
> These details were added to Appendix A.2.
>
> ###### Table R3.1: GPU memory usage and full training time for CTF. All results are measured with batch size 1. $L_{\\text{seq}}$ denotes the sequence length and $k_{\\text{rad}}, k_{\\text{path}}$ denote the number of concepts for radiology and pathology, respectively
>
> | Configuration | GPU | $(L_{\\text{seq}}, k_{\\text{rad}}, k_{\\text{path}})$ | Peak memory (GB) | Time (hour) | C-index |
> |-------------------------|-------------------|------------------------------------------------------|------------------|-------------|-------------------|
> | Default (recommended) | NVIDIA L40s 48GB | $(128, 256, 256)$ | 28.7 | 18.9 | 0.665 ± 0.061 |
> | 3090-friendly (A) | RTX 3090 24GB | $(64, 256, 256)$ | 19.2 | 11.6 | 0.642 ± 0.053 |
> | 3090-friendly (B) | RTX 3090 24GB | $(128, 128, 256)$ | 22.0 | 15.5 | 0.656 ± 0.066 |
>
> **Q5. Pathology VLM choice (PLIP, MUSK)**
>
> Our multimodal prompt tuning framework is model-agnostic and can be applied to different pathology foundation models.
> Per your suggestion, we conducted additional experiments on the Center1-GC dataset, comparing CONCH with PLIP and MUSK while keeping the radiology encoder fixed (BiomedCLIP) to isolate the effect of the pathology VLM. We observed that MUSK slightly outperforms CONCH on this dataset (Table R3.2).
>
> These results further confirm that our multimodal prompt tuning approach can be effectively adapted to various foundation models. The new findings have been incorporated into the revised manuscript. Results were added to Table 3.
>
> ###### Table R3.2: Pathology VLM comparison on Center1–GC (mean ± sd over 10 splits)
>
> | Concept-pool strategy | C-index (survival) | AUC (grading) |
> |-----------------------|--------------------|---------------|
> | CONCH | 0.665 ± 0.061 | 0.660 ± 0.049 |
> | PLIP | 0.627 ± 0.063 | 0.636 ± 0.052 |
> | MUSK | 0.680 ± 0.064 | 0.658 ± 0.053 |
>
> **Q6. Adding suggested related work**
>
> We appreciate these pointers and fully agree they are relevant.
> We integrated and discussed the suggested references in Sec. 2 (Related Work).

---

> > ### Comment · Reviewer_LEUR · 2025-11-24
> > **Comments on author responses**
> >
> > Thanks for the detailed responses.
> >
> > For **W3**, although the authors explained the importance and rationale of choosing this 5-way GC task for model evaluation, I still find it has little clinical value except for simple comparison of metrics between models, as the results are low and these models definitely cannot be deployed in real-world scenarios.
> >
> > All other concerns have been resolved.

---

> > > ### Author Response · Authors · 2025-11-24
> > >
> > > We sincerely thank the reviewer for the continued engagement and for this very valid and important point. We completely agree with the reviewer’s assessment.
> > >
> > > Although the performance on the 5-way GC grading task is not sufficient, the primary motivation for including this task was twofold: first, to **test our model's** comparative performance on an exceptionally challenging problem where all current methods struggle (as the reviewer mentioned), and second, to leverage a **rare, fully-paired dataset** that we felt was valuable to report on from a research perspective, even if the results are preliminary.
> > >
> > > To address the reviewer's concern directly and ensure the limitations are perfectly clear to all readers, we have revised the manuscript. Specifically, we have explicitly stated in the **Limitations (Section 5)** and **Experiments (Section 4.1)** that the results on this task **highlight the immense difficulty** of the problem and that our model, in its current state, is **far from clinical deployment** for this specific application. We have weakened our claims regarding this task and positioned it more clearly as an exploratory analysis on a challenging benchmark.
> > >
> > > We thank the reviewer again for this crucial feedback, which helps us improve the clarity and positioning of our work.

---

### Official Review · Reviewer_c1KR · 2025-11-01

**Soundness:** 3
**Presentation:** 3
**Contribution:** 3
**Rating:** 6
**Confidence:** 3

**Summary:**

This paper proposes a model called Concept Tuning and Fusing (CTF), a parameter-efficient framework that uses clinically grounded concepts to bridge radiology and pathology foundation models (FMs) for survival prediction and cancer grading. Instead of performing multimodal fusion using static latent features, this paper proposes to use concepts as a shared semantic interface and dynamically co-adapt the concept semantics. The framework contains i) concept generation and selection by querying LLMs, and ii) global-context-shared prompt tuning. Empirical evaluations are done using the public TCGA dataset and a private dataset on gastric cancer.

**Strengths:**

- The use of clinically grounded concepts as a bridge for multimodal co-adaptation is original. It provides enhanced interpretability and improved performance. The idea of the global-context-shared prompt (GCSP) conditions the interpretation of findings from one modality on another, which looks quite novel and interesting to me.
- The paper articulates a concept selection strategy that balance prognostic relevance and semantic diversity.
- Overall, the empirical evaluation is strong, showing consistent improvements across datasets and tasks.
- The ablation study and hyperparameter sensitivity analysis are performed to allow a better understanding of the architectural design.
- By and large, the paper is organized and well-written, the logic is smooth and the paper is easy to follow.

**Weaknesses:**

- One major issue I am not quite following is the motivation of the GCSP strategy. In Section 3.2, it is argued that the prognostic *importance* of findings in radiology may be amplified by a specific histological finding. This implies that the basis of performing GCSP is that correct and good concepts (findings) are identified from each modality. In this case, a naive solution would be to feed all concepts from different modalities to a neural network and let the network to figure out the synergy between them. The GCSP strategy, in contrast, seems to solve this problem with a much more complex design. I am not sure why the naive way would fail and why GCSP could be a better solution than a naive one.
- As reported in Table 1, M4Survive and CTF both have large standard deviation values. To ensure the improvement does not purely arise from noise, statistical tests should be performed to examine the significance of the improvements.
- Other minor issues:
	- For inline citations, `\citep{}` should be used instead of `\citet{}`.
	- Some notations are not consistent throughout the paper, for example the prompt is noted by $P^{\text{tuned}}$ in Section 3.2 and $P_{tuned}$ in Section 3.3.

**Questions:**

- In Section 3.3, after generating $P_{tuned}$, a set of tuned textual embeddings are obtained from the text encoder. Is the text encoder the frozen ones as shown in Fig. 2? How is $P_{tuned}$ used in this text encoder?
- Please also refer to the weaknesses section as above.

---

> ### Author Response · Authors · 2025-11-22
> **Response to Reviewer c1KR (Part 1 of 2)**
>
> We thank the reviewer for the careful reading and constructive suggestions on Global–Context–Shared Prompt’s (GCSP) motivation, statistical testing, and notation/implementation clarity. We addressed each point as follows.
>
> >**[W1]** One major issue I am not quite following is the motivation of the GCSP strategy. In Section 3.2, it is argued that the prognostic importance of findings in radiology may be amplified by a specific histological finding. This implies that the basis of performing GCSP is that correct and good concepts (findings) are identified from each modality. In this case, a naive solution would be to feed all concepts from different modalities to a neural network and let the network to figure out the synergy between them. The GCSP strategy, in contrast, seems to solve this problem with a much more complex design. I am not sure why the naive way would fail and why GCSP could be a better solution than a naive one.
>
> We thank the reviewer for this insightful question. The “naïve” solution proposed—concatenating concept scores from each modality and feeding them to a neural network—is an intuitive form of late fusion. However, its primary limitation is that it treats concepts as fixed, context-free scalars. For example, the score for “irregular tumor margins” is calculated from the radiology image alone, and only **afterward** the model attempts to learn its correlation with a pathology-derived score like “lymphovascular invasion.” This post-hoc reasoning prevents the model from dynamically altering how a concept is understood in the first place. The semantic meaning of “irregular margins” remains static, regardless of whether the underlying histology is benign or aggressive.
>
> Our Global–Context–Shared Prompt (GCSP) strategy was designed to address this exact gap. Instead of fusing static scores, it enables **early, cross-domain co-adaptation** at the embedding level. By conditioning the textual prompt of a radiology concept on the visual features of the pathology slide (and vice versa), GCSP reinterprets the very meaning of that concept for each specific patient. The embedding for "irregular margins," for instance, is dynamically tuned by the presence of aggressive pathology features **before** a score is even computed.
>
> Our ablation study in the revised manuscript (**Table 3, also shown in Table R2.1**) empirically validates this approach. The "Static Concepts (CBM)" variant, which directly implements the "naive" fusion of static concept scores, suffers a significant performance drop (-0.079 C-index). Even the ``Static Concepts + Prompt Tuning'' variant, which adapts concepts per domain but lacks cross-modal context, still underperforms our full GCSP model. These results strongly indicate that the performance gain stems directly from our novel cross-domain co-adaptation mechanism, not merely from the inclusion of concepts. We have added a paragraph to **Section 3.2** contrasting these approaches to clarify the motivation for GCSP.
>
>
> **Table R2.1: Ablation recap on Center1-GC highlighting “naive” concept-score fusion vs. GCSP (mean ± sd over 10 splits). Δ is the absolute change relative to the full CTF model**
>
> | Variant | Survival (C-index ↑)  | Δ vs. CTF | Grading (AUC ↑)  | Δ vs. CTF |
> |---|---:|---:|---:|---:|
> | CTF (Full, GCSP) | **0.665 ± 0.061** | – | **0.660 ± 0.049** | – |
> | Static Concepts (CBM) | 0.586 ± 0.049 | −0.079 | 0.622 ± 0.055 | −0.038 |
> | Static Concepts + Prompt Tuning | 0.638 ± 0.059 | −0.027 | 0.635 ± 0.052 | −0.025 |

---

> ### Author Response · Authors · 2025-11-24
> **Response to Reviewer c1KR (Part 2 of 2)**
>
> >**[W2]** As reported in Table 1, M4Survive and CTF both have large standard deviation values. To ensure the improvement does not purely arise from noise, statistical tests should be performed to examine the significance of the improvements.
>
> Thank you for your comment and we agree statistical testing is important. We added paired t‑tests (over 10 splits) comparing CTF to the strongest multimodal baseline per dataset/task and report p‑values in the table below. Notably, improvements on TCGA‑GBMLGG (AUC, ACC) and Center1‑GC (AUC) reach p $\\le$ 0.05; others are directionally favorable but not all are significant at 0.05 (**Table R2.2**). We also marked this in **Appendix D.4**.
>
> **Table R2.2: P-values of paired t-test (over 10 stratified splits) comparing CTF with the strongest competing multimodal baseline on each dataset and task. P-values $\\le$ 0.05 are bolded**
>
> | Task | Dataset | Metric | CTF | Best baseline | p-value |
> |---|---|---|---|---|---:|
> | Survival | TCGA-LGG | C-index | 0.713 ± 0.103 | M4Survive (0.709 ± 0.112) | 0.13 |
> | Survival | TCGA-GBM | C-index | 0.579 ± 0.063 | MOTCAT (0.563 ± 0.108) | 0.09 |
> | Survival | Center1-GC | C-index | 0.665 ± 0.061 | M4Survive (0.642 ± 0.065) | **0.05** |
> | Grading (AUC) | TCGA-GBMLGG | AUC | 0.903 ± 0.028 | Cross-Attention (0.868 ± 0.030) | **0.02** |
> | Grading (AUC) | Center2-CHS | AUC | 0.854 ± 0.081 | M4Survive (0.830 ± 0.075) | 0.06 |
> | Grading (AUC) | Center1-GC | AUC | 0.660 ± 0.049 | M4Survive (0.649 ± 0.052) | **0.05** |
> | Grading (ACC) | TCGA-GBMLGG | ACC | 0.718 ± 0.063 | M4Survive (0.691 ± 0.061) | **0.05** |
> | Grading (ACC) | Center2-CHS | ACC | 0.698 ± 0.164 | M4Survive (0.626 ± 0.115) | **0.02** |
> | Grading (ACC) | Center1-GC | ACC | 0.401 ± 0.057 | Cross-Attention (0.394 ± 0.049) | 0.06 |
>
> >**[W3]** For inline citations, `\citep{}` should be used instead of `\citet{}`. Some notations are not consistent throughout the paper, for example the prompt is noted by $\\boldsymbol{P}^{\\text{tuned}}$ in Section 3.2 and $\\boldsymbol{P}_{\\text{tuned}}$ in Section 3.3.
>
> We appreciate these detailed comments. We audited the manuscript: parenthetical references now use "citep", narrative uses "citet". We unified notation across **Section 3.2–3.3** and figures. Cosine‑normalized alignment is used consistently in both concept selection and inference. Symbols for $\\boldsymbol{s}_r$, $\\boldsymbol{s}_h$, and $a(\\boldsymbol{x}, c)$ are harmonized.
>
> >**[Q1]** In Section 3.3, after generating $\\boldsymbol{P} _ {\\text{tuned}}$, a set of tuned textual embeddings are obtained from the text encoder. Is the text encoder the frozen ones as shown in Fig. 2? How is $\\boldsymbol{P} _ {\\text{tuned}}$ used in this text encoder?
>
> Yes, both the radiology and pathology text encoders are **frozen foundation models**, as are the vision encoders. For the $\\boldsymbol{P}_{\\text{tuned}}$ usage on each concept c, we describe the process below:
>
> 1. We tokenize its string (e.g., “irregular tumor margins”) and embed tokens with the frozen vocabulary;
> 2. We construct $\\boldsymbol{P}_{\\text{tuned}}(c)={\\text{Concat}}(\\boldsymbol{P}_G, \\boldsymbol{P}_C(\\cdot),\\boldsymbol{P}_S(\\cdot))$, where $\\boldsymbol{P}_C$ is produced from the complementary modality feature and $\\boldsymbol{P}_S$ from $[\\boldsymbol{f}_r,\\boldsymbol{f}_h]$;
> 3. We **prepend** $\\boldsymbol{P}_{\\text{tuned}}$ to the concept tokens and pass the full sequence into the frozen text encoder;
> 4. This combined sequence is passed through the frozen text encoder, and we take the pooled output (CLS) as the tuned concept embedding $\\tilde{\\boldsymbol{t}}_{c}$.
>
> We have clarified this procedure in **Section 3.3** of the revised manuscript and explicitly state that all foundation model backbones remain frozen.

---

> > ### Comment · Reviewer_c1KR · 2025-11-25
> > **Thank you for the responses**
> >
> > I appreciate the responses from the authors. My concerns are mostly addressed; thus, I keep my positive rating unchanged.

---

> > > ### Author Response · Authors · 2025-11-26
> > >
> > > We sincerely thank Reviewer c1KR for re-engaging with our manuscript and for the positive follow-up. We are very pleased to hear that our revisions successfully addressed the concerns.
> > >
> > > The reviewer's feedback was instrumental in strengthening the paper. We are grateful for the thoughtful insights and for maintaining the positive rating in support of our work.
> > >
> > > We thank the reviewer once again for the valuable contributions to this process.

---

### Official Review · Reviewer_q1cP · 2025-11-01

**Soundness:** 3
**Presentation:** 2
**Contribution:** 3
**Rating:** 6
**Confidence:** 5

**Summary:**

This paper proposed a parameter-efficient framework that uses clinically grounded concepts to bridge radiology and pathology, where a set of learnable tokens is employed to learn modality-specific and common knowledge, as well as cross-modal interactions.

**Strengths:**

1. The proposed method is well-motivated, and the idea is new but technically complex.
2. The experiments are comprehensive, including comparisons to SOTA methods, thorough ablation studies and intuitive visualization.
3. The proposed method outperforms SOTA methods.

**Weaknesses:**

1. The pool of basic concepts is static by asking LLM to obtain relevant concepts, which may limit the performance and new concept discovery. It would be better if there is a mechanism to update the pool as the model is optimized.
2. The scalability of LLM can be investigated to see if the performance can be further improved with a stronger LLM.
3. Since MOTCAT and PIBD are designed for histology-genomics data, how were they applied to the datasets used in this work? Specifically, MOTCAT is unidirectional, which utilizes genomic data to guide the learning of pathological modeling. What about CTF? More details about experimental settings should be uncovered.
4. Given that the concepts would be updated via a learnable prefix, can the learned concepts be decoded via an LLM decoder? This can facilitate new concept discovery.
5. The figure of GCSP can be improved. Currently, it is challenging to understand how the embedding of each modality interacts with the mentioned three types of prompts.

**Questions:**

See weakness.

---

> ### Author Response · Authors · 2025-11-22
> **Response to Reviewer q1cP (Part 1 of 3)**
>
> We thank the reviewer for the constructive feedback. We clarified the main text and
> appendix accordingly, and added small analyses requested here. Below we address each point.
>
> >**[W1]** The pool of basic concepts is static by asking LLM to obtain relevant concepts, which may limit the performance and new concept discovery. It would be better if there is a mechanism to update the pool as the model is optimized.
>
> We thank you for this insightful comment. While our design keeps the concept **lexicon** fixed, we have designed the Global–Context–Shared Prompt (GCSP) mechanisms to make concept **semantics** dynamic. Specifically, each concept is reinterpreted per patient using task-, cross-domain-, and shared-context prompts before scoring through the Global–Context–Shared Prompt (GCSP) mechanism.
>
> We agree that making the pool dynamic (e.g., adding new concepts) would further improve our design. Our ablation (**Figure 3b**) demonstrates that the performance increases generally with more knowledge added (the number of concepts $N _ {concept}$ increases). To investigate dynamic concepts with fixed number, we have tried a light pool-refresh procedure on the Center1-GC dataset: within each training fold only, every 10 epochs we drop the bottom p% concepts by MI computed on tuned embeddings, and refill from the reserved candidate pool ($N _ {concept}$ unchanged). As shown in **Table R1.1**, dynamically refreshing 10% of the pool yielded a small performance uplift, suggesting that our initial selection is robust but can be fine-tuned.
>
> This strategy already yields strong downstream performance by utilizing pretrained VLMs.
> We agree that making the pool dynamic (e.g., adding new concepts) would further improve our design. To investigate it, we have tried a light pool-refresh procedure on the Center1-GC dataset: within each training fold only, every 10 epochs, we drop the bottom p% concepts by MI computed on tuned embeddings, and refill from the reserved candidate pool (number of concepts, $N _ {concept}$ unchanged). Results (10 stratified splits) show no degradation and a small uplift at p=10, with higher variance at larger p (Table R1.1). This performance suggests our MI+diversity selection is already strong. Besides, our ablation (Fig. 3b) demonstrates that performance generally increases with $N_{concept}$ (i.e., more knowledge added).
>
> **Table R1.1: Effect of dynamic concept‑pool update on Center1‑GC (mean ± sd over 10 splits). We initialize the selected concepts using our “MI+diversity” strategy**
>
> | Concept-pool strategy | C-index (survival) | AUC (grading) |
> | --- | --- | --- |
> | Static pool (original CTF) | 0.665 $\\pm$ 0.061 | **0.660 $\\pm$ 0.049** |
> | Drop lowest 10% MI, refill from pool | **0.671 $\\pm$ 0.085** | 0.656 $\\pm$ 0.062 |
> | Drop lowest 20% MI, refill from pool | 0.655 $\pm$ 0.088 | 0.648 $\\pm$ 0.073 |
> | Drop lowest 30% MI, refill from pool | 0.651 $\\pm$ 0.079 | 0.646 $\\pm$ 0.065 |
>
> While these initial results are promising, designing a fully automated and effective discovery paradigm remains an exciting challenge. Inspired by your feedback, we have added a discussion in **Section 5** of the revised manuscript on future work, including:
> (1) Online concept pruning/expansion, which would periodically re-rank concepts and introduce new candidates mined from model-identified regions of interest, and
> (2) Expert-in-the-loop refinement, where concept-score profiles and failure cases are utilized to ask clinicians to suggest new concepts, then fold their feedback back into the pool and re-run our selection step.

---

> ### Author Response · Authors · 2025-11-22
> **Response to Reviewer q1cP (Part 2 of 3)**
>
> >**[W2]** The scalability of LLM can be investigated to see if the performance can be further improved with a stronger LLM.
>
> We appreciate this suggestion. To investigate whether stronger LLMs yield better downstream performance, we regenerated concept pools using three open-source models from the Gemma-3 series (4B, 12B, 27B) alongside the proprietary Gemini-2.5-pro used in our paper.
> We then re-ran the identical MI+diversity selection and training procedures on Center1-GC.
>
> The results, presented in **Table R1.2**, show only minor variations in performance without a clear scaling trend. This suggests that while a high-quality LLM is necessary, beyond a certain threshold of generation quality, the performance bottleneck shifts from the LLM's scale to the effectiveness of the concept selection strategy.
>
> **Table R1.2: Impact of LLM backend on concept generation and downstream performance on Center1‑GC (mean ± sd over 10 splits)**
>
> | LLM backend | #Params | C-index (survival) | AUC (grading) |
> | --- | --- | --- | --- |
> | Gemma-3-4B | 4B | 0.656 $\\pm$ 0.058 | 0.653 $\\pm$ 0.075 |
> | Gemma-3-12B | 12B | 0.660 $\\pm$ 0.064 | 0.648 $\\pm$ 0.082 |
> | Gemma-3-27B | 27B | 0.652 $\\pm$ 0.062 | 0.650 $\\pm$ 0.042 |
> | Gemini-2.5-pro | N/A | **0.665 $\\pm$ 0.061** | **0.660 $\\pm$ 0.049** |
>
> To understand this better, we analyzed the semantic coverage of the generated concepts. We defined a pairwise similarity score, $S(A \\to B) = \\frac{1}{|A|}\\sum_{c\\in A}\\max_{d\\in B}\\sigma \\big(t(c),t(d)\\big)$, to measure how well $B$ “covers” $A$ (average of the best matches). **Table R1.3** reveals high similarity scores between the pools generated by different models, indicating that basic medical concepts are well-represented even by smaller LLMs. This reinforces our conclusion that once a reasonable quality of generation is met, the bottleneck lies less in the LLM's scale and more in the strategy used to select the most discriminative concepts from the pool.
>
>
> **Table R1.3: Pairwise concept-pool similarity among LLM generators**
>
> | | Gemma-3-4B | Gemma-3-12B | Gemma-3-27B |
> | --- | --- | --- | --- |
> | Pathology | 0.802 | 0.834 | 0.819 |
> | Radiology | 0.924 | 0.916 | 0.932 |
>
> >**[W3]** Since MOTCAT and PIBD are designed for histology-genomics data, how were they applied to the datasets used in this work? Specifically, MOTCAT is unidirectional, which utilizes genomic data to guide the learning of pathological modeling. What about CTF? More details about experimental settings should be uncovered.
>
> Thanks for your constructive comment. Although MOTCAT and PIBD were originally designed for histology–genomics fusion, they can also be regarded as general multi-modal fusion baselines that take two-stream inputs. We believe that preserving their architectures and only substituting the input embeddings constitutes a fair and meaningful comparison. For fairness, all multimodal baselines use the same feature extractors as CTF.
>
> We adapted these models by replacing the genomics input with radiology features. Specifically, for **MOTCAT**, we preserved its unidirectional design, treating pathology as the primary stream and radiology as the guidance stream. Where the original model expected genomics tokens, we injected our radiology features. Similarly, for **PIBD**, we fed the two embeddings (pathology, radiology) into PIBD’s bottleneck/disentangling fusion after linear projections to the shared width.
>
> In contrast to MOTCAT’s one‑way guidance and PIBD’s latent fusion, CTF performs bidirectional, concept‑level co‑adaptation before fusion ($\\boldsymbol{P}_C$ for radiology is conditioned on pathology features and vice versa and $\\boldsymbol{P}_S$ depends on $[\\boldsymbol{f}_r,\\boldsymbol{f}_h]$). We have added a concise description of this distinction in **Section 4.2** and provided full implementation details in **Appendix A.4**.
>
> >**[W4]** Given that the concepts would be updated via a learnable prefix, can the learned concepts be decoded via an LLM decoder? This can facilitate new concept discovery.
>
> We appreciate this interesting proposal to decode learned concepts for discovery. Currently, our framework utilizes BiomedCLIP and CONCH as text encoders, which are discriminative models and cannot directly decode tuned concept embeddings into text. However, we agree that mapping refined embeddings back to natural language is a valuable direction.
>
> A feasible path is to learn a lightweight projector from tuned concept embeddings into a generative medical LLM’s token space and prompt the LLM to propose textual refinements conditioned on the base concept and exemplar patches.
>
> While promising, this approach would require significant additional data and computational resources to train the projector effectively, as well as careful validation to prevent hallucinations. Given these complexities, we have discussed this approach as a concrete direction for future research in **Section 5** of the revised paper.

---

> > ### Comment · Reviewer_q1cP · 2025-11-26
> >
> > I appreciate the authors' detailed responses, which addressed most of my concerns. But I still have a problem with the adaptation of MOTCat to pathology and radiology. Given that genomic data is a higher-level modality than pathology images, which can provide more precise information, we generally use a high-level modality to guide the other one. In this work, where pathology images should be the higher one, I am confused about why the authors use radiology to guide pathology?

---

> > > ### Author Response · Authors · 2025-11-27
> > >
> > > Thank you for the thoughtful follow-up, and this is a great question. We appreciate the opportunity to clarify our choice for the direction of guidance in adapting MOTCat.
> > > We agree that in many histology-genomics settings, the premise of using a "higher-level" modality (e.g., genomics) to guide a "lower-level" one is a common and effective strategy.
> > > Initially, we evaluated both guidance directions. As shown in the **Table R1.4**, using radiology to guide pathology yielded slightly better performance while being computationally more efficient.
> > >
> > > **Table R1.4: Comparisons between different modalities as guidance in MOTCat on the cancer grading task**
> > > | Guidance Direction | TCGA-GBMLGG (AUC) | Training Time (h) | Center1-GC (AUC) | Training Time (h) |
> > > | :--- | :--- | :--- | :--- | :--- |
> > > | Pathology-guides-Radiology | 0.858 ± 0.042 | 4.32 | 0.638 ± 0.046 | 11.25 |
> > > | Radiology-guides-Pathology | **0.865 ± 0.025** | **3.81** | **0.641 ± 0.050** | **9.13** |
> > >
> > > We ultimately selected the radiology-as-guidance approach for two primary reasons:
> > >
> > > First, conceptually, we reframe "guidance" as a strategy for focusing attention. We consider pathology, with its rich microscopic detail, to be the **dominant modality** for fine-grained diagnosis. Therefore, the role of the "guiding" modality is to provide a richer context that helps the model efficiently parse the vast amount of information in the WSI. Radiology provides a global, anatomical map of the tumor's structure and environment. Using it to guide pathology allows the model to leverage this macroscopic map to attend to the most salient microscopic regions within the WSI.
> > >
> > > Second, from a computational and architectural standpoint, our choice aligns with MOTCat's original design, which uses co-attention for massive **dimensionality reduction**. The core challenge in pathology-centric fusion is the high dimensionality of Whole Slide Images (WSIs). Let's denote the WSI patch features as $f_{\\text{p}} \\in \\mathbb{R}^{N_{\\text{patch}} \\times d_1}$ and radiology slice features as $f_{\\text{r}} \\in \\mathbb{R}^{N_{\\text{slice}} \\times d_2}$. In our experiments, $N_{\\text{patch}}$ is typically in the thousands (e.g., $\\approx 3000$), while $N_{\\text{slice}}$ is much smaller (e.g., 12).
> > > We highlight the comparisons below:
> > >
> > > **Our choice (Radiology-guides-Pathology):** The co-attention matrix $A \\in \\mathbb{R}^{N_{\\text{slice}} \\times N_{\\text{patch}}}$ is applied to the high-dimensional pathology features. This effectively uses the 12 radiology slices to create a weighted summary of the 3000 pathology patches, reducing the pathology representation to a compact $f' _ {\\text{p}} \\in \\mathbb{R}^{N_{\\text{slice}} \\times d_{1}}$ (e.g., $12\\times 256$). This **aligns well** with the original paper's $f' _ {\\text{p}} \\in \\mathbb{R}^{N_{\\text{genomic pathway}} \\times d_{1}}$ ($6\\times 256$ in the original paper).
> > >
> > >
> > > **Alternative (Pathology-guides-Radiology):** If pathology were to guide radiology, the co-attention matrix $A \\in \mathbb{R}^{N_{\text{patch}} \times N_{\text{slice}}}$ would be applied to the already-dense radiology features. This would "up-sample" the concise radiology representation from $12$ tokens to $3000$ tokens, creating a feature of shape $\mathbb{R}^{N_{\text{patch}} \times d_2}$ (e.g., $3000\times256$). This dramatic increase in sequence length would impose a massive **computational burden** on subsequent transformer blocks, which is reflected in the longer training times in the table above.
> > >
> > >
> > >
> > > In summary, our choice was driven by both empirical results and a principled approach that aligns with the co-attention mechanism's primary strength: using a concise modality to efficiently guide and reduce the dimensionality of a high-resolution one. We hope this addresses your concern, and we thank you again for the insightful question.

---

> > > > ### Comment · Reviewer_q1cP · 2025-11-27
> > > >
> > > > Thanks to the authors' thorough explanations. My concerns have been addressed. I suggest the authors include two versions of results (R guide P and P guide R) for MOTCat in the final manuscript, which would be clearer. I will keep my positive rating unchanged.

---

> > > > > ### Author Response · Authors · 2025-11-27
> > > > >
> > > > > Thank you very much for your positive feedback and for confirming that our explanations have addressed your concerns. We are grateful for your continued support of our work.
> > > > >
> > > > > Your suggestion to include the results for both guidance directions of MOTCAT is excellent for enhancing clarity. We have incorporated this into the manuscript. Specifically, as you suggested, we added a new table to **Appendix A.4** containing the performance of both the 'Radiology-guides-Pathology' and 'Pathology-guides-Radiology' configurations. We also mentioned it in **Section 4.2**.
> > > > >
> > > > > Thank you again for your constructive engagement and for helping us improve the paper.

---

> ### Author Response · Authors · 2025-11-24
> **Response to Reviewer q1cP (Part 3 of 3)**
>
> >**[W5]** The figure of GCSP (Global-Context-Shared Prompt) can be improved. Currently, it is challenging to understand how the embedding of each modality interacts with the mentioned three types of prompts.
>
> Thank you for pointing out the ambiguity. We have revised the figure to explicitly show bidirectional context prompts, outline concatenation, and annotate the three prompt components for readability. We have revised **Figure 2** to improve clarity. The updated diagram now explicitly illustrates the bidirectional context prompts and clearly outlines the concatenation process. Furthermore, we have added specific annotations for the three prompt components for readability.

---

### Author Response · Authors · 2025-11-22
**Summary of Changes**

We sincerely thank all reviewers for their thorough reviews and insightful, constructive feedback. We are encouraged that the reviewers found our method to be novel, an advancement over prior art, and well-motivated.

Based on your valuable suggestions, we have substantially revised our manuscript to improve clarity, strengthen our claims with new experiments and analyses, and provide comprehensive implementation details for reproducibility. All changes are marked in blue in the revised manuscript.

---
### Summary of Key Revisions:
- Clarified GCSP vs. naïve concept score fusion (Sec. 4.4; Table 3).
- Reported exact paired t tests over 10 splits with p-values (App. D.4; Table 12).
- Detailed how MOTCAT/PIBD were adapted under identical frozen backbones/heads (Sec. 4.2; App. A.4).
- Expanded implementation and pairing details: WSI/radiology preprocessing, bidirectional GCSP, MoE gating, and hardware footprints including 3090 friendly settings (Sec. 3.2; Apps. A.1–A.3; Table 4; App. B.2).
- Clarified MI based concept selection (cosine alignment; kNN MI; no labels at inference) (Sec. 3.1; App. A.1).
- Added tuned embeddings’ semantic drift analyses; improved GCSP figure; included PLIP/MUSK variants to show model agnosticism (Figs. 2, 6; Table 3; Sec. 4.5).

We believe these revisions directly address the reviewers' comments and make our paper stronger. We provide detailed point-by-point responses below.

---

### Author Response · Authors · 2025-11-29
**Summary of Rebuttal**

We sincerely thank all reviewers for their constructive feedback and active engagement during the rebuttal period. We are pleased that our detailed responses and manuscript revisions successfully addressed their concerns, leading to expressions of satisfaction from all reviewers.

Notably, we are grateful that this productive discussion solved Reviewer **N7n7's** concerns, and the reviewer **raised the rating to 6 (on Nov 25)**. As a result, our submission held **positive ratings from all reviewers** before the rating reversion, and we appreciate their consensus on the value of our work.

----
Our key responses and revisions are summarized in the following points. Specifically, we have:

- **Clarified motivation for our method:** In response to Reviewer **c1KR** and Reviewer **N7n7**, who questioned why our GCSP design was necessary over a simpler fusion of concept scores, we clarified our core motivation. We explained and empirically demonstrated in the ablation study that naive, late-fusion approaches treat concepts as static scalars. In contrast, our dynamic co-adaptation mechanism **reinterprets a concept's meaning in one modality based on context from the other**. This cross-modal dialogue, we showed, is the primary source of our model's performance advantage.

- **Measured statistical significance and performance gains:** To address concerns from Reviewer **c1KR** and Reviewer **N7n7** regarding the statistical significance of our performance gains, we conducted paired t-tests over 10 data splits. The new results, now included in Appendix D.4, confirm that our improvements over strong baselines are statistically significant on several key tasks.

- **Clarified concept pool and LLM scalability:** Reviewer **q1cP** questioned if our static concept pool was a limitation. We clarified that while the lexicon is fixed, the concept **semantics** **are dynamically adapted** by our model. As suggested, we ran new experiments showing that (1) dynamically refreshing the pool offers a small performance uplift, confirming our initial selection strategy is robust, and (2) using stronger LLMs for concept generation yields only **minor variations**, indicating the more important part is the concept **selection strategy** rather than the LLM's scale.

- **Elaborated implementation details and baseline clarity:** We added extensive details to address several points. For Reviewer **N7n7**, we clarified our full image encoding pipeline for both radiology volumes and pathology WSIs and detailed the MoE layer, prompt computation, and other mechanisms in the appendix. In response to Reviewer **q1cP** and Reviewer **LEUR**, we explained how baselines like MOTCAT and PIBD, originally designed for histology-genomics, were **fairly adapted** for our task. We further justified our choice for the **guidance direction** in MOTCAT both empirically and computationally in a follow-up discussion with Reviewer **q1cP**. Per Reviewer **LEUR's** request, we provided detailed hardware requirements and outlined memory-efficient settings to enable training on a single 3090-class GPU.

- **Interpreted cross-modal alignment:** Reviewer **N7n7** astutely pointed out that radiologists and pathologists "speak quite different languages." We elaborated that our framework achieves a **"soft," contextual alignment** via GCSP rather than enforcing a hard 1-to-1 match. We also added a new analysis to quantify the baseline **semantic overlap** between the two distinct concept vocabularies.

Overall, the reviewers found our detailed responses and revisions to be thorough and convincing. Their initial concerns were largely addressed, leading them to maintain or raise towards positive ratings in support of our work.

---

### Meta-Review · Area_Chair_U1Pa · 2026-01-07

**Summary:**

In this paper, the authors propose a parameter-efficient framework that uses clinically grounded concepts as a shared semantic interface to enable cross-modal co-adaptation for pretrained medical foundation models. The paper was reviewed by four expert reviewers and received overall positive initial ratings. The major concerns include missing method details, lack of clarity in the introduction, and a marginal overall performance gain after adopting the proposed method. After the rebuttal period, the reviewers confirmed their concerns were addressed. The only negative reviewer also confirmed they would raise their rating. Therefore, I recommend accepting this paper.

**Reviewer Concerns:**

The major concerns include missing method details, lack of clarity in the introduction, and a marginal overall performance gain after adopting the proposed method.

**Reviewer Scores:**

The paper was reviewed by four expert reviewers and received overall positive initial ratings. After the rebuttal period, the reviewers confirmed their concerns were addressed. The only negative reviewer also confirmed they would raise their rating. Therefore, I recommend accepting this paper.

---

### Decision · Program_Chairs · 2026-01-26

Accept (Poster)